# Directed Spectrum Measures Improve Latent Network Models Of Neural Populations

**Neil M. Gallagher**
Department of Neurobiology
Duke University
Durham, NC 27708
`neil.gallagher@duke.edu`

**Kafui Dzirasa**
Howard Hughes Medical Institute
Department of Psychiatry and Behavioral Sciences
Department of Neurobiology
Duke University
Durham, NC 27710
`kafui.dzirasa@duke.edu`

**David Carlson**
Department of Biostatistics and Bioinformatics
Department of Civil and Environmental Engineering
Duke University
Durham, NC 27708
`david.carlson@duke.edu`

## Abstract

Systems neuroscience aims to understand how networks of neurons distributed throughout the brain mediate computational tasks. One popular approach to identify those networks is to first calculate measures of neural activity (e.g. power spectra) from multiple brain regions, and then apply a linear factor model to those measures. Critically, despite the established role of directed communication between brain regions in neural computation, measures of directed communication have been rarely utilized in network estimation because they are incompatible with the implicit assumptions of the linear factor model approach. Here, we develop a novel spectral measure of directed communication called the Directed Spectrum (DS). We prove that it is compatible with the implicit assumptions of linear factor models, and we provide a method to estimate the DS. We demonstrate that latent linear factor models of DS measures better capture underlying brain networks in both simulated and real neural recording data compared to available alternatives. Thus, linear factor models of the Directed Spectrum offer neuroscientists a simple and effective way to explicitly model directed communication in networks of neural populations.

## 1 Introduction

A major goal in neuroscience is to characterize how populations of neurons work together to carry out computational tasks [1]. A well-known example is the biological neural network that identifies low-level visual features such as edges. That network sends signals from the retina to the lateral geniculate nucleus, then from there to the primary visual cortex, then finally to higher cortical visual processing regions [2, 3]. While some biological neural networks are well known, we expect that the vast majority remain undiscovered due to the enormous variety of tasks the brain performs. Many methods have been developed to help discover latent networks of neural populations (i.e. brain networks) [4–6]. A key aspect of such methods is that they should be interpretable, meaning that a neuroscientist must be able to use the model to draw conclusions about brain function [7].

35th Conference on Neural Information Processing Systems (NeurIPS 2021).

Unfortunately, many types of common models in the machine learning literature do not offer this type of interpretability [8].

One class of widely-used models that is considered interpretable for modeling brain networks is Linear Factor Models (LFMs) [9, 10]. This family includes popular methods like principal component analysis (PCA) and independent component analysis (ICA), and are used across almost all modalities in neuroscience to model brain networks. For example, ICA is regularly used in functional magnetic resonance imaging (fMRI) applications to identify latent brain networks such as the default mode network [11, 12]. A common approach in neuroscience is to apply an LFM to some measure or extracted feature from the recorded signal, rather than to the recorded signals themselves, in order to get a model of brain networks that are characterized in the desired measure. For example, power spectra are often calculated from raw electroencephalogram (EEG) recordings before an LFM is applied, yielding brain networks defined by power spectra [13, 14]. In multi-site local field potential (LFP) data, LFMs have been applied to identify latent brain networks that are defined by a cross-spectral covariance matrix [15].

One kind of measure that could be especially informative for defining latent brain networks is directed communication between brain regions. However, LFMs that directly incorporate measures of directed communication are lacking. We believe that this is because existing measures of directed communication, such as Granger causality [16], are incompatible with the implicit assumptions of using LFMs to identify brain networks. This incompatibility between standard measures of directed communication and LFMs is discussed further in Sections 2 and 4. In a previous attempt to capture directed communication within an interpretable brain network model, Gallagher et al. [15] modeled networks in terms of phase shifts in spectral content between neural populations, but it is unclear if such phase shifts are an appropriate proxy for directed transmission of signals, and those models are bottlenecked by significant computational time.

In response to this methodological gap, we introduce a novel measure of directed communication that we will refer to as the Directed Spectrum (DS). The Directed Spectrum estimates directed communication between time series in the frequency domain. We prove that these measures are a linear function of latent brain networks under reasonable assumptions, making them compatible with LFMs in this application. We then demonstrate that an LFM of DS measures recovers latent networks in a simulated dataset where the ground truth networks are known. We compare the performance to several competing directed communication measures, including Spectral Granger Causality [17], and show that using the Directed Spectrum results in significantly higher quality reconstruction of the true brain networks. Finally, we show that latent brain networks identified from real neural data via the Directed Spectrum can decode behaviorally relevant information with much higher fidelity.

## 2 Linear factor models (LFMs) for identifying brain networks

The term *linear factor model* [18] describes any model that seeks to approximate a data vector $\boldsymbol{x}_n$ as,

$$\boldsymbol{x}_n = \sum_{j=1}^{J} Z_n^{(j)} \boldsymbol{x}^{(j)} + \boldsymbol{\epsilon}_n. \tag{1}$$

The vectors $\boldsymbol{x}^{(1)}, \ldots, \boldsymbol{x}^{(J)}$ are the $J$ different latent factor loadings, $\boldsymbol{\epsilon}_n$ is the additive noise term, and $Z_n^{(j)}$ is the activation score of the $j^{th}$ factor in the $n^{th}$ sample. $\boldsymbol{x}_n$ represents a single sample from a larger dataset, $[\boldsymbol{x}_1, \ldots, \boldsymbol{x}_N]$. Many well-known models, such as PCA [19], ICA [20], and nonnegative matrix factorizaton [21], are LFMs. LFMs are frequently used for modeling and discovering brain networks [10]. In that context, it is straightforward to interpret the scores, $Z_n^{(j)}$, as the activation levels for a set of brain networks. Likewise, we can interpret the factor loading $\boldsymbol{x}^{(j)}$ as the observable signature of the $j^{th}$ brain network. LFMs are quite flexible; a wide variety of assumptions can be made regarding the structure of the factors, the scores, and the noise distribution [6, 9, 22]. The fact that LFMs are interpretable and adaptable to most data types has made them a standard model class for studying latent brain networks.

*Note:* For simplicity, the subscript $n$ will be dropped from all variables for the remainder of this document. In cases where a variable is constant over all samples, such as the factor loadings $\boldsymbol{x}^{(j)}$, it will be explicitly stated.

## 2.1 Compatibility between linear factor models and measures of neural activity

It is common to model measures of neural activity recordings rather than modeling the raw recording time-series directly. In this work, we assume that the vector $\boldsymbol{x}$ contains measures calculated from multi-channel time-series, $V = [\boldsymbol{v}_1, \ldots, \boldsymbol{v}_K]^\mathsf{T}$, where $K$ is the number of channels and $\boldsymbol{v}_c \in \mathbb{R}^T$ is the row of $V$ corresponding to a recording from a single channel $c$. The frequency domain representations of these quantities will be marked by a tilde (e.g. $\tilde{v}_c(\omega)$ is the frequency domain representation of $\boldsymbol{v}_c$). A typical choice for $\boldsymbol{x}$ is to convert the observed data into a set of power spectra ($S_{cc}$) for each channel [13, 23],

$$S_{cc}(\omega) \equiv E[|\tilde{v}_c(\omega)|^2]. \tag{2}$$

Modeling these measures in an LFM leads to the latent factor loadings $\boldsymbol{x}^{(j)}$ being described in terms of power spectra associated with each channel. The chosen measures dictate the representation of the discovered networks, so we want to use relevant and easy-to-interpret measures.

When an LFM is used to model latent brain networks, there is an implicit assumption that the measures contained in $\boldsymbol{x}$ are *linear* functions of the latent brain networks. This is seen in (1) where the observed measure is a linear function of the latent factor score $Z^{(j)}$ within some noise tolerance. We let $\mathcal{X}(\cdot)$ represent a function, such as (2), that produces the set of measures from our observed neural data $V$.

**Definition 1.** *We call $\mathcal{X}(\cdot)$ a linear function of latent brain networks if*

$$\mathcal{X}(V) = \mathcal{X}(\textstyle\sum_{j=1}^J Z^{(j)}\Omega^{(j)}) = \textstyle\sum_{j=1}^J Z^{(j)}\mathcal{X}(\Omega^{(j)}), \tag{3}$$

*where $\Omega^{(j)}$ represents the component of the neural data ($V$) that is due to the $j^{th}$ network, normalized to represent a score of 1. In this way, $\mathcal{X}(\Omega^{(j)})$ represents the expected value of the measure if the $j^{th}$ network was active with a score of 1 in isolation.*

It can be shown that power spectra are theoretically consistent with this requirement (see Supplemental Section A), but there are many measures that cannot obey this assumption. For example, the Pearson correlation coefficient and Granger causality are two measures that cannot obey this assumption (see Section 4). Including such measures in LFMs will lead to suboptimal representations of latent brain networks.

## 3 Modeling directed communication in brain networks with LFMs

Measures capturing directed communication between neural populations are rarely included in LFMs despite their frequent use in neuroscience [24–26]. We believe this is because standard measures of directed communication, including Granger causality [16, 17], are not linear functions of brain networks and therefore are poorly modeled by LFMs (see Section 4). Nonetheless, measures of directed communication such as Granger causality are vital for the study of brain networks [27, 28].

Below, we introduce a novel measure of directed communication, which we refer to as the Directed Spectrum (DS). In order to derive the Directed Spectrum, we first build a general time-series model of latent brain networks in Section 3.1, and then define how latent networks in the model combine to produce the observed signal in Section 3.2. The Directed Spectrum is formally defined in Section 3.3. It captures communication in a similar manner to Granger causality while also being compatible with a linear model of brain networks.

### 3.1 Modeling brain networks as independent VAR processes

We outline a model where the output for each latent network is defined by a vector autoregressive (VAR) process. For additional background on VAR models, see Supplemental Section B. The model outlined below represents a general framework for understanding what the DS measures capture, even though inferring it directly would be quite challenging. We begin by assuming that we have $J$ latent brain networks that each generate vector timeseries outputs. Rather than slicing by channel as in Section 2, we define the output series as $V^{(j)} = [\boldsymbol{v}_1^{(j)}, \ldots, \boldsymbol{v}_T^{(j)}]$, where $\boldsymbol{v}_t^{(j)} \in \mathbb{R}^K$ is the column of $V^{(j)}$ representing signal for all channels at time $t$. The output series is associated with a VAR process,

$$\boldsymbol{v}_t^{(j)} = \textstyle\sum_{\tau=1}^{p_j} A_\tau^{(j)} \boldsymbol{v}_{(t-\tau)}^{(j)} + \boldsymbol{\sigma}_t^{(j)}, \qquad \boldsymbol{\sigma}_t^{(j)} \sim \mathcal{N}(0, Z^{(j)}\Sigma^{(j)}). \tag{4}$$

We refer to $\boldsymbol{\sigma}_t^{(j)}$ as the *innovations* in the $j^{th}$ network at time $t$, and assume that they are drawn *iid* from a zero mean Gaussian distribution. Each network has a single covariance structure over all samples ($\Sigma^{(j)}$) that is scaled by an activation score ($Z^{(j)}$) to define the innovation distribution of the current sample. The set of autocovariance matrices for the $j^{th}$ network $\{A_1^{(j)}, \ldots, A_{p_j}^{(j)}\}$ are also assumed to be constant over all samples. The innovation terms represent new signal introduced into the network that cannot be explained by the past network outputs $\{\boldsymbol{v}_{(t-1)}^{(j)}, \ldots, \boldsymbol{v}_{(t-p_j)}^{(j)}\}$. We can represent this network in the frequency domain given standard assumptions regarding the stability of the VAR model [29],

$$\tilde{\boldsymbol{v}}^{(j)}(\omega) = H^{(j)}(\omega)\tilde{\boldsymbol{\sigma}}^{(j)}(\omega), \quad H^{(j)}(\omega) \equiv \left(I - \sum_{\tau=1}^{p_j} A_\tau^{(j)} e^{-i\tau\omega}\right)^{-1}, \quad 0 \leq \omega \leq 2\pi, \quad (5)$$

where $\tilde{\boldsymbol{v}}^{(j)}(\omega) \in \mathbb{R}^K$ and $\tilde{\boldsymbol{\sigma}}^{(j)}(\omega) \in \mathbb{R}^K$ are the frequency domain representations of the series $\boldsymbol{v}_1^{(j)}, \ldots, \boldsymbol{v}_T^{(j)}$ and $\boldsymbol{\sigma}_1^{(j)}, \ldots, \boldsymbol{\sigma}_T^{(j)}$, respectively. The network transfer matrix ($H^{(j)}(\omega)$) is assumed to be constant over all samples.

### 3.2 Modeling observed signal as a superposition of latent transmitted signals

In order to complete our model of latent brain networks, we must relate it to the observed signal. We model each observed sample by a VAR in the frequency domain, analogous to the way we modeled latent network outputs,

$$\tilde{\boldsymbol{v}}(\omega) = H(\omega)\tilde{\boldsymbol{\sigma}}(\omega). \tag{6}$$

We assume that the $K$ channels in our data are partitioned into a set of non-overlapping groups, $\mathcal{G} = \{\boldsymbol{b}, \boldsymbol{c}, \boldsymbol{d}, \ldots\}$. For example, each group can be a single channel to model all inter-channel relationships, or could be all channels within a given brain region. We partition the observed data and the transfer matrix $H(\omega)$ and innovations $\tilde{\boldsymbol{\sigma}}(\omega)$ into these groups,

$$
\tilde{\boldsymbol{v}}(\omega) = \begin{bmatrix} \tilde{\boldsymbol{v}}_{\boldsymbol{b}}(\omega) \\ \tilde{\boldsymbol{v}}_{\boldsymbol{c}}(\omega) \\ \tilde{\boldsymbol{v}}_{\boldsymbol{d}}(\omega) \\ \vdots \end{bmatrix}, \quad
H(\omega) = \begin{bmatrix} H_{\boldsymbol{bb}}(\omega) & H_{\boldsymbol{bc}}(\omega) & H_{\boldsymbol{bd}}(\omega) \\ H_{\boldsymbol{cb}}(\omega) & H_{\boldsymbol{cc}}(\omega) & H_{\boldsymbol{cd}}(\omega) & \cdots \\ H_{\boldsymbol{db}}(\omega) & H_{\boldsymbol{dc}}(\omega) & H_{\boldsymbol{dd}}(\omega) \\ & & \vdots \end{bmatrix},
$$
$$
\tilde{\boldsymbol{\sigma}}(\omega) = \begin{bmatrix} \tilde{\boldsymbol{\sigma}}_{\boldsymbol{b}}(\omega) \\ \tilde{\boldsymbol{\sigma}}_{\boldsymbol{c}}(\omega) \\ \tilde{\boldsymbol{\sigma}}_{\boldsymbol{d}}(\omega) \\ \vdots \end{bmatrix}, \quad
\Sigma = \begin{bmatrix} \Sigma_{\boldsymbol{bb}} & \Sigma_{\boldsymbol{bc}} & \Sigma_{\boldsymbol{bd}} \\ \Sigma_{\boldsymbol{cb}} & \Sigma_{\boldsymbol{cc}} & \Sigma_{\boldsymbol{cd}} & \cdots \\ \Sigma_{\boldsymbol{db}} & \Sigma_{\boldsymbol{dc}} & \Sigma_{\boldsymbol{dd}} \\ & & \vdots \end{bmatrix}.
\tag{7}
$$

We assume that the networks outputs and corresponding VAR parameters are partitioned in the same way. The ordering of these groups is arbitrary. Without loss of generality, we will focus on modeling communication between groups $\boldsymbol{b}$ and $\boldsymbol{c}$.

We examine how our VAR models represent communication between groups by noting that

$$\tilde{\boldsymbol{v}}_{\boldsymbol{c}}(\omega) = \sum_{\boldsymbol{g} \in \mathcal{G}} H_{\boldsymbol{cg}}(\omega)\tilde{\boldsymbol{\sigma}}_{\boldsymbol{g}}(\omega). \tag{8}$$

This shows that $\tilde{\boldsymbol{v}}_{\boldsymbol{c}}(\omega)$ is a sum of contributions from each group $\boldsymbol{g}$, including the self-contribution from $\boldsymbol{c}$. If the innovations in $\boldsymbol{b}$ and $\boldsymbol{c}$ are independent, then $H_{\boldsymbol{cb}}(\omega)\tilde{\boldsymbol{\sigma}}_{\boldsymbol{b}}(\omega)$ would unambiguously represent the signal in $\boldsymbol{c}$ that can be attributed to innovations $\boldsymbol{b}$. If the innovations in $\boldsymbol{b}$ and $\boldsymbol{c}$ are not independent, then there is ambiguity regarding how to assign 'responsibility' for the observed signal in $\boldsymbol{c}$. Specifically, the component of the innovations in $\boldsymbol{b}$ that is correlated with innovations in $\boldsymbol{c}$ is not necessary to explain the observed signal in $\boldsymbol{c}$ in expectation, since the innovations in $\boldsymbol{c}$ are sufficient to explain this part of the observed signal. We will assume that only the uncorrelated component of the innovations in $\boldsymbol{b}$ contribute to the observed signal in $\boldsymbol{c}$.

**Definition 2.** *The transmitted signal represents the contribution of the innovations in $\boldsymbol{b}$ to the observed signal in $\boldsymbol{c}$*

$$\mathcal{TS}_{\boldsymbol{b} \to \boldsymbol{c}}(\omega) = \begin{cases} H_{\boldsymbol{cb}}(\omega)\left(\tilde{\boldsymbol{\sigma}}_{\boldsymbol{b}}(\omega) - \Sigma_{\boldsymbol{bc}}\Sigma_{\boldsymbol{cc}}^{-1}\tilde{\boldsymbol{\sigma}}_{\boldsymbol{c}}(\omega)\right), & \boldsymbol{b} \neq \boldsymbol{c} \\ \sum_{\boldsymbol{g} \in \mathcal{G}} H_{\boldsymbol{cg}}(\omega)\Sigma_{\boldsymbol{gc}}\Sigma_{\boldsymbol{cc}}^{-1}\tilde{\boldsymbol{\sigma}}_{\boldsymbol{c}}(\omega), & \boldsymbol{b} = \boldsymbol{c}. \end{cases} \tag{9}$$

By restricting $\mathcal{TS}_{b \to c}(\omega)$ to only convey uncorrelated innovations, we construct a more conservative estimate of the effect of $b$ on $c$. This is not the only choice that could be made; for example, we could use the full innovations, but we believe that using the uncorrelated innovations provides the cleanest interpretation.

We complete our model of the observed signal $\tilde{v}(\omega)$ with the following two assumptions:

**Assumption 1.** *The innovations associated with any network are independent of the innovations in all other networks.*

**Assumption 2.** *The transmitted signals of the observed data are the sum of the corresponding transmitted signals for the latent networks,*

$$\mathcal{TS}_{b \to c}(\omega) = \sum_{j=1}^{J} \mathcal{TS}_{b \to c}^{(j)}(\omega), \tag{10}$$

*where the transmitted signal is defined for the latent networks as*

$$\mathcal{TS}_{b \to c}^{(j)}(\omega) = \begin{cases} H_{cb}^{(j)}(\omega)\left(\tilde{\sigma}_b^{(j)}(\omega) - \Sigma_{bc}^{(j)}\Sigma_{cc}^{(j)^{-1}}\tilde{\sigma}_{c,n}^{(j)}(\omega)\right), & b \neq c \\ \sum_{g \in \mathcal{G}} H_{cg}^{(j)}(\omega)\Sigma_{gc}^{(j)}\Sigma_{cc}^{(j)^{-1}}\tilde{\sigma}_c^{(j)}(\omega), & b = c \end{cases}. \tag{11}$$

Assumption 1 can be thought of as enforcing the independence of inputs to each network. Assumption 2 is equivalent to the following two statements: 1) the propagation of signal within each network is independent of propagation in each other network, and 2) the observed outputs occur in some medium where simultaneously occurring phenomena obey the laws of superposition, which is a property of many neural activity measurement modalities (e.g., electrical potentials, fluorescence). By combining these two assumptions with (8), we get the result that the observed data are a superposition of the network outputs,

$$\tilde{v}_c(\omega) = \sum_{g \in \mathcal{G}} H_{cg}(\omega)\tilde{\sigma}_g(\omega) \qquad = \sum_{g \in \mathcal{G}} \mathcal{TS}_{g \to c}(\omega) \tag{12}$$

$$= \sum_{g \in \mathcal{G}} \sum_{j=1}^{J} \mathcal{TS}_{g \to c}^{(j)}(\omega) \quad = \sum_{g \in \mathcal{G}} \sum_{j=1}^{J} H_{cg}^{(j)}(\omega)\tilde{\sigma}_g^{(j)}(\omega) = \sum_{j=1}^{J} \tilde{v}_c^{(j)}(\omega). \tag{13}$$

Note in (12) and (13), summing the *transmitted signals* to $c$ over all source groups, including $c$, causes the conditioning terms in our definitions of the *transmitted signal* to cancel out. Because the relationship above does not contain a noise term, non-physiological sources of signal would simply be represented as additional networks in the model. This completes our model of the relationship between latent brain networks and observed signals.

## 3.3  The Directed Spectrum

The model defined above provides a framework for determining whether a measure of neural recordings can be considered a linear function of latent brain networks.

**Definition 3.** *We define our measure, the Directed Spectrum (DS), as the second moment of the transmitted signal,*

$$\mathcal{DS}_{b \to c}(\omega) \equiv \mathbb{E}\left[\widetilde{\mathcal{TS}}_{b \to c}(\omega)\widetilde{\mathcal{TS}}_{b \to c}^{*}(\omega)\right] = H_{cb}(\omega)\Sigma_{b|c}H_{cb}^{*}(\omega), \tag{14}$$

$$\Sigma_{b|c} = \Sigma_{bb} - \Sigma_{bc}\Sigma_{cc}^{-1}\Sigma_{bc}^{*},$$

*where $\Sigma_{b|c}$ represents the innovation variance for $b$ conditioned on the innovations in $c$.*

The Directed Spectrum corresponds to the portion of the power spectrum for $c$ that is explained by signal that originated in $b$ (see Supplemental Section C for more details). It is a linear function of the latent networks when data is appropriately modeled as described above (for proof, see Supplemental Section D).

The Directed Spectrum can be estimated efficiently by fitting a VAR model to the observed data or via factorization of the cross-spectral matrix associated with the observed data [24, 30, 31]. Additional algorithmic details can be found in Supplemental Section F.1. These DS measures can be calculated according to (14) for data partitioned into any number of groups greater than one. In some situations, it is desirable to calculate the DS values separately for each pair of groups, by estimating $H(\omega)$ and $\Sigma$ using reduced models that contains only two groups per model. We refer to this method as the *Pairwise Directed Spectrum* (PDS). One benefit of the PDS is that it

simplifies the handling of missing channels from the data, since the PDS for the non-missing groups of channels can still be calculated without any adjustments. Code for calculating the Directed Spectrum and Pairwise Directed Spectrum is provided in the supplemental material (MATLAB) and at `https://github.com/neil-gallagher/directed-spectrum` (Python).

## 4 Related measures of directed communication

The Directed Spectrum is a necessary development because similar measures typically used in neuroscience are not linear functions of latent brain networks. We first examine Granger causality, which measures the degree to which one signal or group of signals helps predict future values of another signal or group [16]. Spectral Granger causality is a modification that separates Granger causality into components associated with individual frequencies [17]. It is a leading spectral measure of directed communication in neuroscience [24, 25], and we view it as conceptually the closest measure to the Directed Spectrum. The (unconditional) spectral Granger causality from $\boldsymbol{b}$ to $\boldsymbol{c}$ can be defined in terms of the Directed Spectrum,

$$\boldsymbol{f}_{\boldsymbol{b}\to\boldsymbol{c}}(\omega) = \ln\left(\frac{|S_{\boldsymbol{cc}}(\omega)|}{|S_{\boldsymbol{cc}}(\omega) - \mathcal{DS}_{\boldsymbol{b}\to\boldsymbol{c}}(\omega)|}\right). \tag{15}$$

It is also possible to calculate a conditional spectral Granger causality that accounts for all other groups in the recording before assigning influence from $\boldsymbol{b}$ to $\boldsymbol{c}$ [32]. Neither form of spectral Granger causality is a linear function of the network models described in Section 3. To see this, consider the case where only the $j^{th}$ network is present without any other signals being added to it. Note that changing $Z^{(j)}$ would then have no impact on $\boldsymbol{f}_{\boldsymbol{b}\to\boldsymbol{c}}(\omega)$ because both the numerator and denominator in (15) would scale equally with $Z^{(j)}$. Standard (non-spectral) Granger causality is equivalent to a scaled integral over all frequencies of the spectral Granger causality [17], meaning that it also would remain constant as $Z^{(j)}$ changes.

There are several other spectral measures of directed communication used in neuroscience, including phase slope index [33], partial directed coherence [34], and the directed transfer function [35]. Each of these can not be a linear function of latent brain networks under reasonable assumptions. We offer the following brief explanations here, with more details in Supplemental Section E. The phase slope index cannot capture bidirectional transmission of signal (all signal is considered unidirectional) [36]. Partial directed coherence and the directed transfer function are both based in VAR models like the Directed Spectrum, but they rely solely on the transfer matrix properties and do not use the innovations. As such, they are scale-invariant and are not linear functions of the brain networks.

There are a number of other non-spectral measures of directed communication used in neuroscience [25, 26]. The cross-correlation function between activity in different neural populations has been used to identify directed communication between neuronal populations [37]. In these applications it has been used to identify the lag associated with unidirectional transmission between populations, and is considered unreliable in situations involving bidirectional communication [25]. Transfer entropy provides an information theoretic measure of directed communication. In our application transfer entropy suffers from the same problem as Granger causality, where scaling the activation of a single network results in the same value [38, 39]. We discuss methods for measuring directed communication between neural populations further in Supplemental Section E.

One final method worth mentioning here is dynamic causal modeling (DCM) [40]. DCM produces a generative model to explain observed neural activity based on latent connectivity. Directed connectivity is represented by parameters of the model, which contrasts with our application where measures of directed connectivity are calculated first before being used as features in an LFM. DCM has even been extended in order to represent spectral content in neural signals [41, 42]. One substantial difference between DCM and the approach we suggest here is that DCM is not designed to segregate the directed connectivity out into subnetworks in an unsupervised manner. Instead, DCM is typically used to identify differences in connectivity between two known conditions [43].

## 5 Directed Spectrum improves identification of latent networks

To test the effectiveness of the Directed Spectrum and LFMs, we generated a dataset of simulated latent networks. The dataset contains 10,000 independent recordings, each containing 5 channels

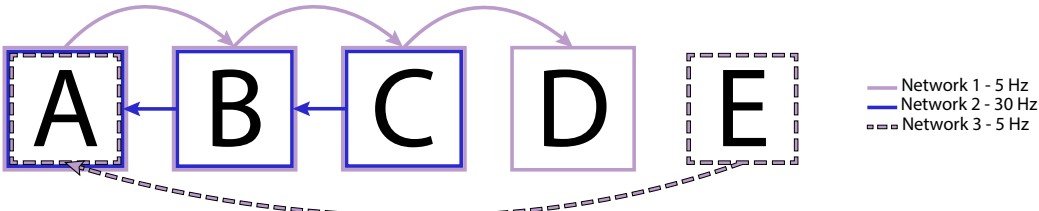

Figure 1: Graphical representation of the simulated latent networks. The five channels are represented by the letters A, B, C, D, and E. A box around a channel indicates the corresponding network induces oscillations in that channel. Arrows indicate propagation of signal at some delay. Networks 1 and 3 have a predominant frequency of 5 Hz (both in purple), and Network 2 has a predominant frequency of 30 Hz (in blue).

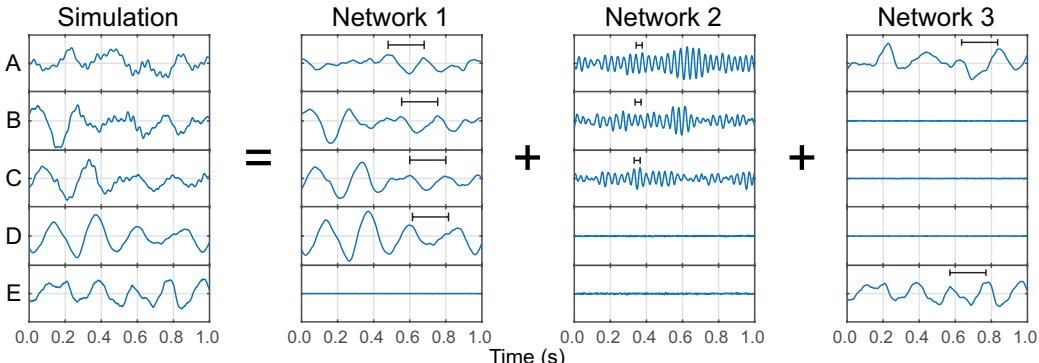

Figure 2: Simulated recordings are a sum of latent network contributions. The "observed" recording (left) is a sum of signals generated by each of the three networks. Each panel contains 5 signals associated with the 5 regions over the same one second period. Black scale bars indicate the period for oscillations associated with each network, and are placed to show that the distance between peaks in the signal is approximately one period.

sampled for five seconds at 500 Hz. The "observed" recordings were generated as a sum of contributions from three independent networks. Each network was associated with a vector autoregressive (VAR) model that generated network contributions as a different random draw for each recording with parameters that were fixed over all recordings. Each VAR model was designed to generate a particular pattern of oscillations and delayed directional transmission of signals. See Figure 1 for the network properties and Figure 2 for a visualization of the superposition. See Supplemental Section F.2 for additional details on the design of those VAR models.

We generated a model of the latent networks in the simulated dataset by applying an LFM to DS measures. We calculated the Directed Spectrum for frequencies between 1 Hz and 50 Hz at 1 Hz intervals, between all directed pairs of channels. A non-negative matrix factorization model (i.e. LFM) was trained using the Itakura-Saito (IS) divergence loss, with an L1 penalty on the factor loadings and scores, and multiplicative update steps for optimization [22]. The IS divergence loss was chosen because it is related to the gamma distribution and is a more appropriate loss for power than mean squared error loss. After training, the non-negative matrix factorization (NMF) factors were re-ordered to correspond to the matching network in the simulation, based on maximizing the average Spearman's correlation between the estimated and true network activation scores. For comparison, we repeated this process with the Pairwise Directed Spectrum (PDS), unconditional spectral Granger causality (GC) [17], conditional spectral Granger causality (cGC) [32], phase-slope index (PSI) [33], directed transfer function (DTF) [35], and partial directed coherence (PDC) [34] features substituted for the DS features. We also use the difference between (unconditional) Granger causality values in either direction as suggested by Roebroeck et al. [44] as an additional comparison method. Because both this difference and PSI represent the direction of shared information with sign (i.e. positive/negative), we set negative values to zero and leave the opposite direction as positive

Table 1: Spearman's correlation between latent network activation estimates and true activation scores. GC: unconditional Granger causality; cGC: conditional Granger Causality; GCdiff: difference between Granger causality directions; PSI: phase slope index; DTF: directed transfer function; PDC: partial directed coherence; DS: Directed Spectrum; PDS: Pairwise Directed Spectrum. Values in *[brackets]* represent the 95% confidence interval [45].

| Measure | Network 1 | Network 2 | Network 3 |
|---|---|---|---|
| GC | 0.485 *[0.468, 0.502]* | 0.442 *[0.424, 0.459]* | 0.281 *[0.261, 0.300]* |
| cGC | 0.475 *[0.458, 0.491]* | 0.293 *[0.274, 0.312]* | 0.127 *[0.108, 0.147]* |
| GCdiff | 0.554 *[0.539, 0.569]* | 0.501 *[0.484, 0.517]* | 0.444 *[0.427, 0.462]* |
| PSI | 0.387 *[0.368, 0.405]* | 0.135 *[0.115, 0.155]* | 0.248 *[0.229, 0.268]* |
| DTF | 0.426 *[0.408, 0.443]* | 0.131 *[0.111, 0.151]* | 0.542 *[0.526, 0.557]* |
| PDC | 0.560 *[0.545, 0.575]* | 0.154 *[0.134, 0.174]* | 0.445 *[0.427, 0.462]* |
| **DS** | 0.920 *[0.916, 0.923]* | 0.905 *[0.901, 0.909]* | 0.927 *[0.924, 0.930]* |
| **PDS** | 0.908 *[0.904, 0.912]* | 0.917 *[0.913, 0.920]* | 0.916 *[0.913, 0.920]* |

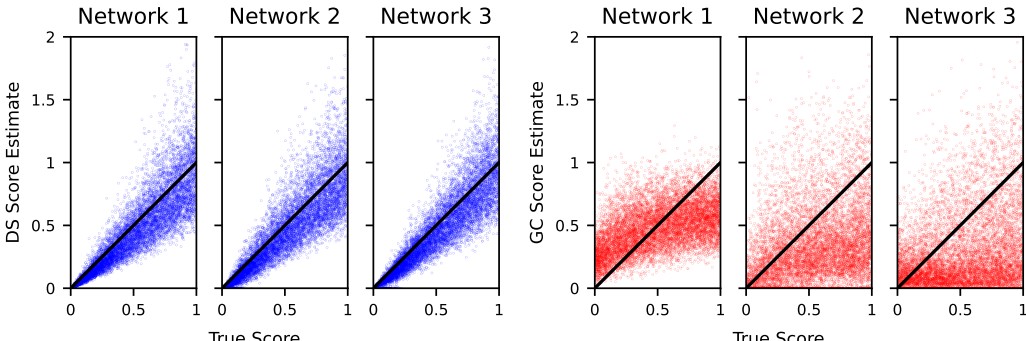

Figure 3: True vs. estimated network scores. The estimated scores for the model trained using DS features are plotted in blue, the estimated scores for a model trained on unconditional Granger causality features are plotted in red. In all plots, a scaling factor was applied to the estimated scores to minimize the mean squared error. The line in black demonstrates the expected trend for a model that perfectly recovers the latent network scores.

for both of those comparison methods; this allowed us to continue to use NMF to identify latent networks.

We evaluated how well each model recovered the underlying network activation scores using Spearman's correlation. The model of DS features performed significantly better in this regard than the "non-linear" features (see Table 1). The PDS model also performed significantly better than the "non-linear" models, and gave comparable results to the DS model. We have visualized the estimated scores along with the true scores for each window for the DS features and unconditional Granger causality features in Figure 3. We see that the score estimates of the DS model are much more tightly spread around the true score values, with the relationship between true scores and estimated scores being very weak in the model of GC features.

We also tested how robust these results were to violations of the model assumptions described in Section 3 and to variations in recording window length. Violations of the model assumptions did reduce model performance somewhat, but the DS features still performed better than the comparison methods. Shorter window lengths reduced performance in all models, but again the DS models performed better at all window lengths. Full details and results for these claims are in Supplemental Sections G and H.

Finally, we tested whether the models produced accurate and interpretable representations of the true latent networks from the dataset. The model of DS features successfully recovers all directional influences of each network in the corresponding factors (see Figures 4, S1). Spurious detection of network components was limited to connections between true nodes in the network at the frequency associated with the network, and these spurious influences were small relative to the true detected

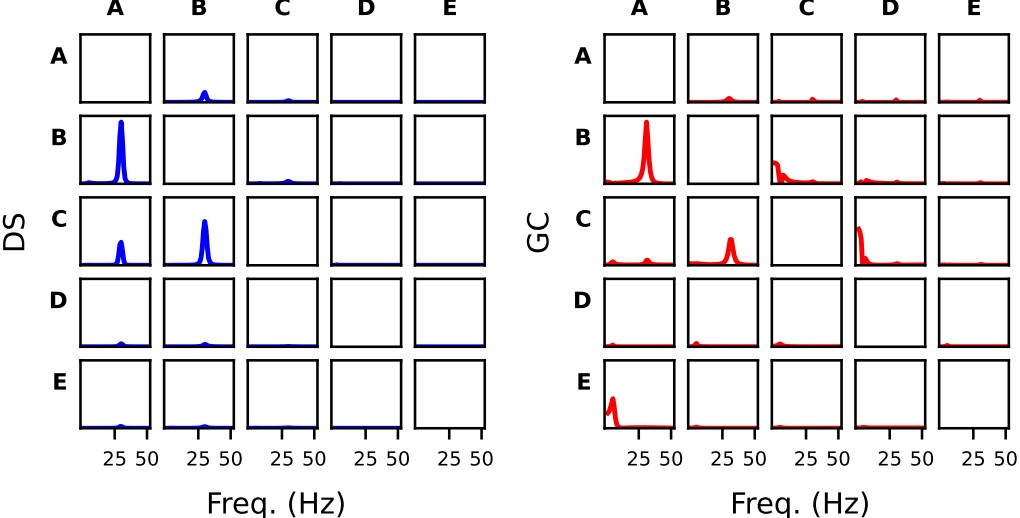

Figure 4: Estimated directional influences in Network 2 for a model of DS features (blue) and unconditional Granger causality features (red). Within each grid, a plot corresponds to signal that is being transmitted from the channel listed on the left to the channel listed above.

influences. While the model of GC features did also recover all directional influences of the networks underlying the simulated data, it incorrectly spread the directional influences associated with each network out over multiple estimated networks. This resulted in the GC model identifying more spurious influences with greater relative amplitudes.

## 6 Directed Spectrum improves brain state decoding of neural recordings

We next tested how effectively networks identified by the Directed Spectrum decode behaviorally relevant variables from a dataset of neural recordings, as a proxy for determining whether they reflect "real" networks. The dataset consists of local field potentials simultaneously recorded from 11 brain regions in 26 mice (originally published in [27]). In each recording session local field potentials were recorded from a mouse while it was exposed to three different behavioral contexts of successively increasing stress levels: resting in the home cage, exploring an open field, and a tail suspension test. The tail suspension test is commonly used to investigate learned helplessness [46].

In order to train networks that could be used to decode behavioral context, we first divided the dataset into time windows with a duration of 1 second. For each time window, we calculated the Directed Spectrum for all directed pairs of brain regions, for frequencies from 1 to 56 Hz at 1 Hz intervals. A nonnegative matrix factorization model with L1 regularization was trained using the IS divergence objective [47] in order to identify latent brain networks from the DS measures. A multinomial logistic regression classifier with L1 regularization was then applied to the latent factor scores for decoding the behavioral context. In order to obtain estimates of the spread of the decoding performance and choose optimal hyperparameter values, a 5-fold nested cross-validation procedure was used, where each mouse was associated with only one split. The average one-vs-all area under the reciever operating characteristic curve (AUC) across all three behavioral contexts was used as the evaluation metric. The hyperparameters tuned during cross-validation were the number of NMF factors (20, 40, 80), NMF regularization strength (1000, 100, 10, 0), and logistic regression regularization strength (10, 1). Similar to the experiment described in Section 5, the procedure was repeated using the Pairwise Directed Spectrum (PDS) and the comparison measures listed in that section. The one comparison method that was not used here is conditional Granger causality; in our tests, obtaining stable conditional spectral Granger causality estimates on our rodent LFP data via the methods outlined by Barnett and Seth [24] requires too much computation time to be practical for this application.

Our training procedure resulted in a one-vs-all AUC for each of the behavioral contexts, for each of the 5 splits, for each of the measure types (see Table 2). We performed a 2-factor repeated measures

ANOVA and determined that there was a significant difference associated with the measure types ($F = 92.9$, $p < .001$). A Tukey's HSD post-hoc test revealed that there was not a significant difference between models of the pairwise and non-pairwise versions of DS ($p = .90$), but that both performed significantly better than models of the other measures ($p \leq .001$ for all).

Table 2: Behavioral Context Decoding Performance. The columns 'HC AUC', 'OF AUC', and 'TS AUC' report the mean and standard error of the one-vs-all AUC across 5 splits for the homecage, open field, and tail suspension behavioral contexts, respectively. The 'Mean AUC' column reports the average across the mean AUCs reported for each behavioral context.

| Measure | *Mean AUC* | HC AUC | OF AUC | TS AUC |
|---------|------------|--------|--------|--------|
| GC | 0.828 | $0.825 \pm 0.019$ | $0.824 \pm 0.022$ | $0.835 \pm 0.010$ |
| GCdiff | 0.795 | $0.795 \pm 0.013$ | $0.798 \pm 0.023$ | $0.791 \pm 0.007$ |
| PSI | 0.674 | $0.676 \pm 0.014$ | $0.684 \pm 0.015$ | $0.661 \pm 0.016$ |
| DTF | 0.755 | $0.774 \pm 0.017$ | $0.774 \pm 0.015$ | $0.717 \pm 0.022$ |
| PDC | 0.717 | $0.733 \pm 0.021$ | $0.778 \pm 0.014$ | $0.639 \pm 0.008$ |
| **DS** | **0.908** | $0.894 \pm 0.016$ | $0.916 \pm 0.012$ | $0.915 \pm 0.007$ |
| **PDS** | **0.919** | $0.909 \pm 0.014$ | $0.915 \pm 0.014$ | $0.932 \pm 0.005$ |

## 7 Discussion

We have shown that our novel measure of directed communication, the Directed Spectrum (DS), is a linear function of latent brain networks under reasonable assumptions, and so is compatible with LFMs for characterizing latent brain networks. We saw drastically improved recovery of networks in simulated data with a known ground truth. In real neural recordings, the Directed Spectrum improved decoding of neurally relevant environmental variables. These results demonstrate that the Directed Spectrum allows for more accurate and interpretable models of latent networks defined by directed communication.

A limitation that could be explored in future work is that the Directed Spectrum assumes that the network states are stationary over a single time window (see Supplemental Section B). This limitation is common to almost all methods that assess spectral content of neural activity. Because of this, the Directed Spectrum is only theoretically grounded when applied to time windows that are shorter than the time expected for substantial change to be occur in the latent brain state. In practice, time window lengths of one to five seconds have been considered a relatively stable period in studies of emotional processing [48, 49].

We have only explored LFMs here, but a variety of nonlinear latent factor models exist for modeling latent brain networks [5, 50]. Nonlinear models are especially useful in applications where predictive performance is the only priority, such as brain-computer interfaces. When the primary goal is to drive scientific understanding of the brain, nonlinear models are less desirable because it is challenging to relate the parameters of such models to relevant conclusions about brain function [8]. We believe that LFMs and the Directed Spectrum are an appropriate, efficient, and reproducible baseline approach in both scientific and prediction-based applications, while noting that other more expressive models may lead to even better predictive performance at the cost of interpretability.

The development of the Directed Spectrum provides a straightforward way to generate linear models of latent brain networks defined by directed communication. We view this as an important advancement because directed communication between neural populations is a critical component of the way neuroscientists understand brain networks. Furthermore, the Directed Spectrum is likely useful for studying latent networks in many other fields as well (e.g. latent networks of directed internet traffic). We view the major strength of the Directed Spectrum is that it enhances the expressiveness of LFMs by allowing them to accurately model directed communication within networks. Thus, we have expanded the capabilities of LFMs while retaining the level of interpretability that make them attractive models in practice.

## Acknowledgments and Disclosure of Funding

We thank the anonymous reviewers for their suggestions to evaluate how robust the methods are to assumption violations, which led to the additional experiments of Supplemental Sections G and H.

Research reported in this manuscript was supported by the National Institute of Biomedical Imaging and Bioengineering and the National Institute of Mental Health through the National Institutes of Health BRAIN Initiative under Award Number R01EB026937, the National Institute of Mental Health of the National Institutes of Health under Award Numbers R01MH125430 and R01MH120158, and the Hope for Depression Research Foundation.

The content is solely the responsibility of the authors and does not necessarily represent the official views of the National Institutes of Health.

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
