

(a) Network 1

(b) Network 3

Figure S1: Estimated Networks 1 & 3 from linear factor models of DS (Top) and Granger causality (Bottom) for simulated data experiment. Each panel shows a grid of DS or Granger causality (GC) features associated with the indicated network estimate. Within each grid, a plot corresponds to signal that is being transmitted from the channel listed on the left to the channel listed at the top. See Fig. 1 for a description of the true networks.

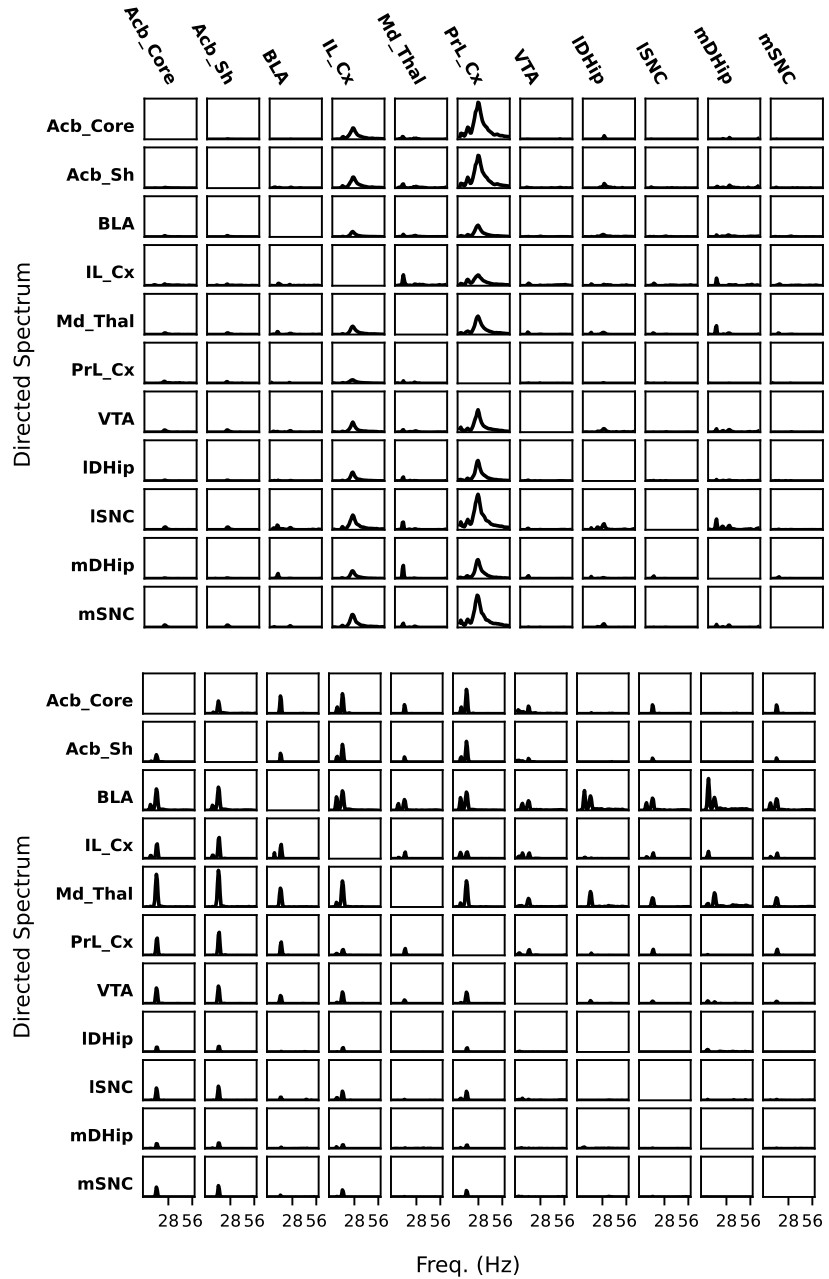

Figure S2: Two example networks from local field potential experiment. Each subplot represents the DS from the region listed on the left to the region listed on top. *Brain region key:* Acb_Core = Nucleus Accumbens Core; Acb_Sh = Nucleus Accumbens; BLA = Basolateral Amygdala; IL_Cx = Infralimbic Cortex; Md_Thal = Mediodorsal nucleus of the Thalamus; PrL_Cx = Prelimbic Cortex; VTA = Ventral Tegmental Area; lDHip = lateral Dorsal Hippocampus; lSNC = lateral Substantia Nigra Pars Compacta; mDHip = medial Dorsal Hippocampus; mSNC = medial Substantia Nigra Pars Compacta

## A  The power spectrum is linear with respect to underlying brain networks

Power spectra are reasonable to model using a linear factor model because they satisfy Definition 1 under reasonable assumptions. We will use $S_{cc}(\omega)$ to refer to the spectral power of the signal $\boldsymbol{v}_c(t)$ at frequency $\omega$, and $\tilde{\boldsymbol{v}}_c(\omega)$ to refer to the frequency domain representation of $\boldsymbol{v}_c(t)$ at $\omega$. This gives the relationship,

$$S_{cc}(\omega) \equiv E[|\tilde{\boldsymbol{v}}_c(\omega)|^2]. \tag{16}$$

Using this definition, we show that Assumption 1 and 2 are sufficient to have the power spectrum be linear with respect to the same networks,

$$S_{cc}(\omega) = E\left[\tilde{\boldsymbol{v}}_c(\omega)^* \tilde{\boldsymbol{v}}_c(\omega)\right] \tag{17}$$

$$= E\left[\left(\sum_{j=1}^{J} \tilde{\boldsymbol{v}}_c^{(j)}(\omega)\right)^* \left(\sum_{l=1}^{L} \tilde{\boldsymbol{v}}_c^{(l)}(\omega)\right)\right] \tag{18}$$

$$= E\left[\sum_{j=1}^{J} \tilde{\boldsymbol{v}}_c^{(j)}(\omega)^* \tilde{\boldsymbol{v}}_c^{(j)}(\omega)\right] \tag{19}$$

$$= \sum_{j=1}^{J} Z^{(j)} S_{cc}^{(j)}(\omega), \tag{20}$$

where $S_{cc}^{(j)}(\omega)$ is the power spectrum of the signal that would be generated if only the $j^{th}$ network were present with an activation score of 1. The equality in (18) follows from (12) and (13). Assumption 1 gives (19). Finally, the network model outlined by (4) gives (20). For notational convenience we have dropped the sample index $n$. This result implies that it is completely reasonable to model spectral power and DS features jointly in the same linear factor model.

## B  Vector autoregressive models

A very common assumption when dealing with neural timeseries recordings is that the recorded signal within each window $n$ is approximately stationary, and is appropriately modeled as a VAR process [24, 29, 34, 35]. For local field potential and electroencephalogram (EEG) recordings, it has been shown that this assumption is reasonable as long as the duration of the samples is relatively short, normally on the order of a few seconds or less [14, 51].

Vector autoregressive (VAR) models are a very common and effective way to understand the spatiotemporal properties of a stationary multivariate timeseries. A VAR model of the vector timeseries $\boldsymbol{v}_1, \boldsymbol{v}_2, \dots, \boldsymbol{v}_T$ represents the vector signal at each time point, $\boldsymbol{v}_t$, as a sum of components determined by past values of the time series and an innovation component,

$$\boldsymbol{v}_t = \sum_{\tau=1}^{p} A_\tau \boldsymbol{v}_{t-\tau} + \boldsymbol{\sigma}_t. \tag{21}$$

In (21), $p$ is referred to as the model order, and defines the number of previous time points that directly contribute to $\boldsymbol{v}_t$. The autoregressive matrices, $A_1, A_2, \dots, A_p$, define the how the previous signal values influence $\boldsymbol{v}_t$. The innovation term, $\boldsymbol{\sigma}_t$, represents the component of $\boldsymbol{v}_t$ that is not accounted for by the past $p$ time points, and is assumed to be independent and identically distributed (*iid*) for each time point. It is typically assumed that the innovations $\boldsymbol{\sigma}_t$ are generated by a zero-mean Gaussian process with covariance $\Sigma$, $\boldsymbol{\sigma}_t \sim \mathcal{N}(0, \Sigma)$.

Another value that is important for understanding the properties of a given VAR model is the transfer matrix, $H(\omega)$, which is fully defined by the autoregressive matrices,

$$H(\omega) \equiv \left(I - \sum_{\tau=1}^{p} A_\tau e^{-i\tau\omega}\right)^{-1}, \qquad 0 \le \omega \le 2\pi. \tag{22}$$

The transfer matrix describes how the innovations give rise to the observed signal in the frequency domain.

$$\tilde{\boldsymbol{v}}(\omega) = H(\omega)\tilde{\boldsymbol{\sigma}}(\omega). \tag{23}$$

For further reference on vector autoregressive models see [29].

## C Relationship between the Directed Spectrum and the cross-spectral matrix

The cross-spectral matrix is an important descriptor used in the analysis of vector timeseries [25, 52]. It is defined as the expectation of the outer product of the frequency representation of a signal with itself,

$$S_{\boldsymbol{v}}(\omega) \equiv E\left[\tilde{\boldsymbol{v}}(\omega)\tilde{\boldsymbol{v}}(\omega)^*\right].\tag{24}$$

The relationship in (23) implies that the cross-spectral matrix associated with $\tilde{\boldsymbol{v}}(\omega)$ can be decomposed into a quadratic expression involving $H(\omega)$ and $\Sigma$.

$$S_{\boldsymbol{v}}(\omega) = H(\omega)\Sigma H^*(\omega)\tag{25}$$

We drop the $\omega$ parameter from $\Sigma$ because the covariance matrix does not depend on frequency under the assumption that the signal is generated by a vector autoregressive process and $\boldsymbol{\sigma}_t$ are *iid* Gaussian terms. This quadratic term describes how the innovation signal, $\boldsymbol{\sigma}_t$, propagates to create the spectral properties of the signal $\boldsymbol{v}_t$. The cross-spectral matrix can be factorized into a *unique* set of VAR parameters $H(\omega)$ and $\Sigma$ [53]. If there are groups within the vector $\tilde{\boldsymbol{v}}(\omega)$ this factorization of the cross-spectral can be partitioned following the scheme given in Section 3.2,

$$\begin{bmatrix} S_{\boldsymbol{bb}}(\omega) & S_{\boldsymbol{bc}}(\omega) & S_{\boldsymbol{bd}}(\omega) \\ S_{\boldsymbol{cb}}(\omega) & S_{\boldsymbol{cc}}(\omega) & S_{\boldsymbol{cd}}(\omega) & \cdots \\ S_{\boldsymbol{db}}(\omega) & S_{\boldsymbol{dc}}(\omega) & S_{\boldsymbol{dd}}(\omega) \\ & \vdots \end{bmatrix} =$$

$$\begin{bmatrix} H_{\boldsymbol{bb}}(\omega) & H_{\boldsymbol{bc}}(\omega) & H_{\boldsymbol{bd}}(\omega) \\ H_{\boldsymbol{cb}}(\omega) & H_{\boldsymbol{cc}}(\omega) & H_{\boldsymbol{cd}}(\omega) & \cdots \\ H_{\boldsymbol{db}}(\omega) & H_{\boldsymbol{dc}}(\omega) & H_{\boldsymbol{dd}}(\omega) \\ & \vdots \end{bmatrix} \begin{bmatrix} \Sigma_{\boldsymbol{bb}} & \Sigma_{\boldsymbol{bc}} & \Sigma_{\boldsymbol{bd}} \\ \Sigma_{\boldsymbol{bc}}^T & \Sigma_{\boldsymbol{cc}} & \Sigma_{\boldsymbol{cd}} & \cdots \\ \Sigma_{\boldsymbol{bd}}^T & \Sigma_{\boldsymbol{cd}}^T & \Sigma_{\boldsymbol{dd}} \\ & \vdots \end{bmatrix} \begin{bmatrix} H_{\boldsymbol{bb}}(\omega)^* & H_{\boldsymbol{cb}}(\omega)^* & H_{\boldsymbol{db}}(\omega)^* \\ H_{\boldsymbol{bc}}(\omega)^* & H_{\boldsymbol{cc}}(\omega)^* & H_{\boldsymbol{dc}}(\omega)^* & \cdots \\ H_{\boldsymbol{bd}}(\omega)^* & H_{\boldsymbol{cd}}(\omega)^* & H_{\boldsymbol{dd}}(\omega)^* \\ & \vdots \end{bmatrix}.$$
$$\tag{26}$$

To simplify our initial exploration of the relationship between the Directed Spectrum and the cross-spectral matrix, we assume that the innovation terms for each group are independent, which causes the off-diagonal blocks of the innovation covariance matrix in (26) to be zero. In doing so, we get that the power spectral density ($S_{\boldsymbol{cc}}(\omega)$) can be split into a separate term for the innovation associated with each group,

$$S_{\boldsymbol{cc}}(\omega) = H_{\boldsymbol{cc}}(\omega)\Sigma_{\boldsymbol{cc}}(\omega)H_{\boldsymbol{cc}}(\omega)^* + H_{\boldsymbol{cb}}(\omega)\Sigma_{\boldsymbol{bb}}(\omega)H_{\boldsymbol{cb}}(\omega)^* + H_{\boldsymbol{cd}}(\omega)\Sigma_{\boldsymbol{dd}}(\omega)H_{\boldsymbol{cd}}(\omega)^* + \cdots$$
$$\tag{27}$$

The first term may be interpreted as the *intrinsic* component of the power spectrum in $\boldsymbol{c}$. The second term may be interpreted as the component of power in $\boldsymbol{c}$ that is predicted by the innovations in $\boldsymbol{b}$. In this case where the innovations terms for each group are independent, that second term is equivalent to the Directed Spectrum from $\boldsymbol{b}$ to $\boldsymbol{c}$.

### C.1 Correlated innovation terms

When the innovation terms are correlated, it is conceptually the same as the previous derivation after a rotation. Regardless, we include the derivation here on two channels for completeness. Multi-channel extensions are straightforward.

Spectral factorization allows us to decompose the CPSD into a special quadratic expression [53],

$$\begin{bmatrix} S_{cc}(\omega) & S_{cb}(\omega) \\ S_{cb}^*(\omega) & S_{bb}(\omega) \end{bmatrix} = \begin{bmatrix} H_{cc}(\omega) & H_{cb}(\omega) \\ H_{bc}(\omega) & H_{bb}(\omega) \end{bmatrix} \begin{bmatrix} \Sigma_{cc} & \Sigma_{cb} \\ \Sigma_{cb}^T & \Sigma_{bb} \end{bmatrix} \begin{bmatrix} H_{cc}^*(\omega) & H_{bc}^*(\omega) \\ H_{cb}^*(\omega) & H_{bb}^*(\omega) \end{bmatrix}.\tag{28}$$

From this, we see that the power spectral density ($S_{cc}(\omega)$) can be split into multiple terms.

$$S_{cc}(\omega) = H_{cc}(\omega)\Sigma_{cc}H_{cc}^*(\omega) + H_{cb}(\omega)\Sigma_{bb}H_{cb}^*(\omega) + 2\Re(H_{cc}(\omega)\Sigma_{cb}H_{cb}^*(\omega)).\tag{29}$$

As before, the first term may be interpreted as the *intrinsic* component of the power in $c$. If the innovations are uncorrelated, the third term would drop out and the second term may be interpreted as the component of power in $c$ that is due to the innovations in $b$ according to the VAR model. If

the innovations are correlated, we may still find a separation of $S_{cc}(\omega)$ into intrinsic and causal components by first applying a linear transformation,

$$\begin{bmatrix} c_t \\ \widetilde{b}_t \end{bmatrix} = U \begin{bmatrix} c_t \\ b_t \end{bmatrix}, \quad U = \begin{bmatrix} 1 & 0 \\ -\Sigma_{bc}\Sigma_{cc}^{-1} & 1 \end{bmatrix}. \tag{30}$$

In this new space, $\Sigma_{c\widetilde{b}} = 0$, and we can split $S_{cc}(\omega)$ into intrinsic and causal components that can still be defined in terms of the original signals $c$ and $b$,

$$S_{cc}(\omega) = H_{cc}(\omega)\Sigma_{cc}H_{cc}^*(\omega) + H_{c\widetilde{b}}(\omega)\Sigma_{\widetilde{b}\widetilde{b}}H_{c\widetilde{b}}^*(\omega), \tag{31}$$

$$= H_{cc}(\omega)\Sigma_{cc}H_{cc}^*(\omega) + H_{cb}(\omega)\Sigma_{b|c}H_{cb}^*(\omega), \tag{32}$$

$$\Sigma_{b|c} = \Sigma_{bb} - \Sigma_{bc}\Sigma_{cc}^{-1}\Sigma_{bc}^*. \tag{33}$$

The causal component of power $(H_{cb}(\omega)\Sigma_{b|c}H_{cb}^*)$ is exactly the Directed Spectrum as defined in Section 3.3.

## D  The Directed Spectrum is a linear function of latent brain networks

**Theorem S1.** *The Directed Spectrum is a linear function of latent brain networks.*

*Proof.* We show that the Directed Spectrum is a linear function of the latent brain networks defined in Section 3 by starting with the definition from (14),

$$\mathcal{DS}_{b\to c}(\omega) = \mathbb{E}\left[\mathcal{TS}_{b\to c}(\omega)\mathcal{TS}_{b\to c}^{\,*}(\omega)\right], \tag{34}$$

$$= \mathbb{E}\left[\left(\sum_{j=1}^{J}\mathcal{TS}_{b\to c}^{(j)}(\omega)\right)\left(\sum_{j'=1}^{J}\mathcal{TS}_{b\to c}^{*(j')}(\omega)\right)\right], \tag{35}$$

$$= \mathbb{E}\left[\left(\sum_{j=1}^{J}H_{cb}^{(j)}(\omega)\left(\tilde{\boldsymbol{\sigma}}_b^{(j)}(\omega) - \Sigma_{bc}^{(j)}\Sigma_{cc}^{(j)-1}\tilde{\boldsymbol{\sigma}}_c^{(j)}(\omega)\right)\right)\right.$$
$$\left.\left(\sum_{j'=1}^{J}\left(\tilde{\boldsymbol{\sigma}}_b^{(j')}(\omega) - \Sigma_{bc}^{(j')}\Sigma_{cc}^{(j')-1}\tilde{\boldsymbol{\sigma}}_c^{(j')}(\omega)\right)^* H_{cb}^{(j')\,*}(\omega)\right)\right], \tag{36}$$

$$= \sum_{j=1}^{J}H_{cb}^{(j)}(\omega)\mathbb{E}\left[\left(\tilde{\boldsymbol{\sigma}}_b^{(j)}(\omega) - \Sigma_{bc}^{(j)}\Sigma_{cc}^{(j)-1}\tilde{\boldsymbol{\sigma}}_c^{(j)}(\omega)\right)\right.$$
$$\left.\left(\tilde{\boldsymbol{\sigma}}_b^{(j)}(\omega) - \Sigma_{bc}^{(j)}\Sigma_{cc}^{(j)-1}\tilde{\boldsymbol{\sigma}}_c^{(j)}(\omega)\right)^*\right]H_{cb}^{(j)\,*}(\omega), \tag{37}$$

$$= \sum_{j=1}^{J}H_{cb}^{(j)}(\omega)\left(Z^{(j)}\Sigma_{b|c}^{(j)}\right)H_{cb}^{(j)\,*}(\omega), \tag{38}$$

$$= \sum_{j=1}^{J}Z^{(j)}\left(H_{cb}^{(j)}(\omega)\Sigma_{b|c}^{(j)}H_{cb}^{(j)\,*}(\omega)\right), \tag{39}$$

$$= \sum_{j=1}^{J}Z^{(j)}\mathcal{DS}_{b\to c}^{(j)}(\omega). \tag{40}$$

Assumption 2 gives us (35). Substituting the definition of *transmitted signal* for networks in (11) gives (36). Assumption 1 then gives (37). The definitions of variance and the conditional variance matrix are then applied to give (38). We define the Directed Spectrum associated with the $j^{th}$ network in a homologous manner to (14).

$$\mathcal{DS}_{b\to c}^{(j)}(\omega) \equiv H_{cb}^{(j)}(\omega)\Sigma_{b|c}^{(j)}H_{cb}^{(j)\,*}(\omega) \tag{41}$$

This creates an equivalency between (34) and (40) in a way that satisfies Definition 1, completing our proof. ∎

# E Established measures of directed communication are not linear functions of latent brain networks

## E.1 Phase slope index

The phase slope index (PSI) was introduced as a measure for estimating the flow of information between channels in vector timeseries [33]. It is defined as

$$\tilde{\Psi}_{bc} = \mathcal{I}\left(\sum_{f \in F} C^*_{bc}(\omega)C_{bc}(\omega + \delta\omega)\right),\tag{42}$$

where $\mathcal{I}(\cdot)$ represent the imaginary component of the expression in parentheses, $F$ is a group of sequential frequencies for which the PSI is being calculated, and $\delta\omega$ is the frequency resolution of the recording. $C_{bc}(\omega)$ is the coherency for the channels $b$ and $c$ at frequency $\omega$,

$$C_{bc}(\omega) = \frac{S_{bc}(\omega)}{\sqrt{S_{bb}(\omega)S_{cc}(\omega)}},\tag{43}$$

where $S_{bc}(\omega)$ is an element of the cross-spectral matrix (see Supplemental Section C). PSI is an approximate estimate of the change in the phase of the frequency domain representation of the data as a function of frequency. A positive value of $\tilde{\Psi}_{bc}$ indicates that information predominantly flows from channel $b$ to $c$, while a negative value indicates information flows from $c$ to $b$. In this way PSI is symmetric, $\tilde{\Psi}_{bc} = -\tilde{\Psi}_{cb}$.

It has been shown that the phase-slope index does not accurately model bidirectional flow of information [36]. This precludes it from being able to accurately represent many real brain networks in practice. In theory, PSI is also not a linear function of latent brain networks as outlined by Definition 1 and Section 3. To see this, consider the case where only the $j^{th}$ latent network is present. We note that the power spectrum and other elements of the cross-spectral matrix should both scale linearly with the network activation $Z^{(j)}$ (for more details see Supplemental Section A). This means the coherency $C_{bc}(\omega)$, and therefore also the PSI $\tilde{\Psi}_{bc}$, would be invariant to changes in $Z^{(j)}$, so they cannot be linear functions of $Z^{(j)}$.

## E.2 Directed transfer function and partial directed coherence

The directed transfer function was developed to measure information flow in multichannel electroencephalogram (EEG) recordings [35]. It is defined in terms of the transfer matrix for a VAR model of the observed data,

$$\mathcal{DTF}_{bc}(\omega) = \frac{H_{bc}(\omega)}{\sqrt{\sum_{g \in \mathcal{G}} |H_{bg}(\omega)|^2}},\tag{44}$$

where $\mathcal{G}$ represents the set of all channels in the dataset. The partial directed coherence is a complementary measure of information flow [34],

$$\mathcal{PDC}_{bc}(\omega) = \frac{\bar{A}_{bc}(\omega)}{\sqrt{\sum_{g \in \mathcal{G}} |\bar{A}_{gc}(\omega)|^2}},\tag{45}$$

$$\bar{A}_{bc}(\omega) = H_{bc}(\omega)^{-1} = \left(I - \sum_{\tau=1}^{p_j} A^{(j)}_\tau e^{-i\tau\omega}\right).\tag{46}$$

Both the directed transfer function and partial directed coherence do not account for the innovation term $\tilde{\boldsymbol{\sigma}}(\omega)$ in the VAR model model of the observed data. This rules both measures out of being linear functions of the latent brain networks defined in Section 3, since they do not scale with the strength of the network.

## E.3 Spectral Granger causality

Spectral Granger causality measures the degree to which one group of signals $\boldsymbol{c}$ helps to predict oscillatory patterns in another group of signals $\boldsymbol{d}$, over a range of frequencies [17]. It is derived from

the theory of vector autoregressive (VAR) models (see Section B). Specifically, spectral Granger causality takes advantage of the relationship between the cross-power spectral density and the VAR model for the corresponding data that was defined by Wilson [53],

$$\boldsymbol{f}_{\boldsymbol{c}\to\boldsymbol{d}}(\omega) \equiv \ln\left(\frac{|S_{\boldsymbol{dd}}(\omega)|}{\left|S_{\boldsymbol{dd}}(\omega) - H_{\boldsymbol{dc}}(\omega)\Sigma_{\boldsymbol{c}|\boldsymbol{d}}H_{\boldsymbol{dc}}(\omega)^*\right|}\right). \tag{47}$$

Here $\boldsymbol{f}_{\boldsymbol{c}\to\boldsymbol{d}}(\omega)$ represents the spectral Granger causality from $\boldsymbol{c}$ to $\boldsymbol{d}$ at frequency $\omega$.

Spectral Granger causality is related to the standard time-domain definition of Granger causality as,

$$\frac{1}{2\pi}\int_0^{2\pi} \boldsymbol{f}_{\boldsymbol{c}\to\boldsymbol{d}}(\omega)\,d\omega \leq \mathcal{F}_{\boldsymbol{c}\to\boldsymbol{d}}, \tag{48}$$

with equality when $\left|A_{\boldsymbol{cc}}(\omega) - \Sigma_{\boldsymbol{cd}}\Sigma_{\boldsymbol{dd}}^{-1}A_{\boldsymbol{dc}}(\omega)\right| \neq 0$ is satisfied in the range $0 < \omega \leq 2\pi$ [17]. In this way, one can think of $\boldsymbol{f}_{\boldsymbol{c}\to\boldsymbol{d}}(\omega)$ as the spectral decomposition of the standard Granger causality $\mathcal{F}_{\boldsymbol{c}\to\boldsymbol{d}}$.

There is a conditional definition of spectral Granger causality that accounts for the effects of another group of recordings $\boldsymbol{g}$ before evaluating the impact of $\boldsymbol{c}$ on the prediction of $\boldsymbol{d}$ [17]. In this case, the effects of the recordings $\boldsymbol{g}$ are considered using a VAR model.

$$\begin{pmatrix}\boldsymbol{d}_t \\ \boldsymbol{g}_t\end{pmatrix} = \sum_{\tau=1}^p \begin{pmatrix}A_{\boldsymbol{dd},\tau} & A_{\boldsymbol{dg},\tau} \\ A_{\boldsymbol{gd},\tau} & A_{\boldsymbol{gg},\tau}\end{pmatrix}\begin{pmatrix}\boldsymbol{d}_{t-\tau} \\ \boldsymbol{g}_{t-\tau}\end{pmatrix} + \begin{pmatrix}\boldsymbol{d}_t^\dagger \\ \boldsymbol{g}_t^\dagger\end{pmatrix} \tag{49}$$

The conditional spectral Granger causality $\boldsymbol{f}_{\boldsymbol{c}\to\boldsymbol{d}|\boldsymbol{g}}(\omega)$ is then defined as an unconditional spectral Granger causality for a different pair of variables.

$$\boldsymbol{f}_{\boldsymbol{c}\to\boldsymbol{d}|\boldsymbol{g}}(\omega) \equiv \boldsymbol{f}_{\boldsymbol{c}\oplus\boldsymbol{g}^\dagger\to\boldsymbol{d}^\dagger}(\omega) \tag{50}$$

with $\boldsymbol{c}_t \oplus \boldsymbol{g}_t^\dagger \equiv \begin{pmatrix}\boldsymbol{c}_t \\ \boldsymbol{g}_t^\dagger\end{pmatrix}$. Just as in the unconditional case, the conditional spectral Granger causality can be considered the spectral decomposition of time-domain conditional Granger causality.

$$\frac{1}{2\pi}\int_0^{2\pi} \boldsymbol{f}_{\boldsymbol{c}\to\boldsymbol{d}|\boldsymbol{g}}(\omega)\,d\omega \leq \mathcal{F}_{\boldsymbol{c}\to\boldsymbol{d}|\boldsymbol{g}} \tag{51}$$

In Section 4, we discussed why all these forms of Granger causality can not be linear functions of latent brain networks.

### E.4 Other measures

A full review of all the available measures of directed communication is beyond the scope of this work. We refer readers to reviews by Bastos and Schoffelen [25] and Wang et al. [54] for such information. We have chosen measures for comparison here that we believe provide a good representation of the most important measures of directed communication used in neuroscience.

## F Detailed methods

### F.1 Estimation of the Directed Spectrum

The directed spectrum can be estimated for a pair of channels within time window by a simple application of (14) once $H(\omega)$ and $\Sigma$ are known. Two well defined methods for estimating $H(\omega)$ and $\Sigma$ from multi-channel timeseries data are spectral factorization [30, 31] and directly modeling the timeseries as a VAR process [24]. For the purpose of this work we followed the spectral factorization method of Wilson [30] to estimate $H(\omega)$ and $\Sigma$.

The first step in the estimation process calculates an estimate of the cross-power spectral density matrix associated with each window. This was done in MATLAB with the CPSD function, which uses Welch's method [55] of averaging multiple shorter time windows to estimate the cross-power spectral density matrix. In both experiments, we used 0.2 $s$ long rectangular windows with 0.175 $s$ of overlap to generate our estimates. Wilson's spectral factorization is an iterative algorithm for

factorizing a cross-power spectral density into the quadratic term $H\Sigma H^*$ [30]. The algorithm has quadratic convergence properties. In this manner, $H(\omega)$ and $\Sigma$ were estimated once for each time window, then the Directed Spectrum was calculated for each directed pair of channel groups in the dataset.

To estimate the Pairwise Directed Spectrum for all directed pairs of channel groups, we first estimated a separate cross-power spectral density for each (undirected) pair of channel groups. We then applied Wilson's factorization to each individual cross-power spectral density, resulting in a separate estimate of $H(\omega)$ and $\Sigma$ for each pair of channel groups. Those values of $H(\omega)$ and $\Sigma$ were then applied in (14) to calculate the Pairwise Directed Spectrum for the corresponding pair. Code for performing these calculations in MATLAB is provided in the supplemental material and in Python is provided at `https://github.com/neil-gallagher/directed-spectrum`.

## F.2  Simulated data generation

Each of the three networks underlying the simulated dataset is defined by a VAR process over the five regions. The parameters of each VAR process are designed to induce oscillations in the regions indicated by Figure 1 at the associated frequencies by setting a single pair of complex conjugate poles for the process. Signals were transmitted between the indicated regions by adding a scaled copy of signal in the sending region to signal in the receiving region at a fixed delay that was specific to each network. The innovations covariance associated with each network was defined by $\Sigma^{(j)} = I + (R^{(j)} + R^{(j)T})/10$, where the elements of $R^{(j)} \in \mathcal{R}^{5\times5}$ are random samples from a standard normal distribution.

Table S1: Simulated network VAR parameters.

| Network | intra-region poles | inter-region delay (ms) | inter-region signal scaling |
|---------|--------------------|-----------------------|----------------------------|
| 1 | $0.98e^{\pm i2\pi\frac{5}{500}}$ | 20 | 0.003 |
| 2 | $0.9e^{\pm i2\pi\frac{30}{500}}$ | 6 | 0.02 |
| 3 | $0.98e^{\pm i2\pi\frac{5}{500}}$ | 20 | 0.003 |

For each simulated recording three activation scores were sampled from a uniform $[0, 1]$ distribution for the networks. The innovation covariance matrix of each network was scaled by the corresponding activation score before sampling a series from the associated VAR model. The three series for each recording were then added together to produce the simulated data for that recording. Code for simulating this data and carrying out the rest of the experiment outlined in Section 5 is provided with the supplemental material for this work.

## F.3  Model training setup

After calculating all of the features listed in Tables 1 and 2 for each dataset, we trained a separate non-negative matrix factorization (NMF) model to represent each set of features. The Directed Spectrum and Pairwise Directed Spectrum features were normalized by the associated frequency to account for differences in spectral across frequencies in local field potential data [56]. The other features tested are invariant to changes in scale, and therefore did not require such normalization. Each NMF model was trained using the NMF function from *scikit-learn*'s DECOMPOSITION module. All models were initialized using nonnegative double singular value decomposition, with zeros filled in with small random values, and were trained using a multiplicative update solver.

### F.3.1  Simulated data experiment

All NMF models were trained with three components, with an L1 regularization strength of 0.1 applied to the components of the model. The KL divergence loss was used to train the NMF model of the phase slope index and Granger causality differences due to increased sparsity in those features. All other models were trained using the Itakura-Saito divergence loss [22].

### F.3.2   Local field potential experiment

For the local field potential dataset, we evaluated each set of features based on decoding performance via a nested cross validation strategy. The mice in the dataset were divided evenly into five splits. For each set of features and split, a final NMF model was trained on the remaining four splits, followed by a multinomial logistic regression classifier trained to classify behavioral context from the latent factor scores from the NMF model of those four splits. The performance for decoding behavioral context was evaluated on the remaining split, giving five total one-vs-all "area under the receiver operating characteristic curve" (AUC) values for each behavioral context and feature set.

All NMF models were trained with L1 regularization on all elements of the model. NMF models trained in the "inner" cross validation loop had a lower tolerance on the stopping condition (1e-3), compared to the tolerance of 1e-4 used in training the final NMF model for each testing split. Each logistic regression model was trained via a SAGA solver with L1 regularization on the model weights.

### F.4   Compute resources

Computational tasks described above were run on a consumer desktop machine with an Intel i9-7980XE CPU and 64GB of RAM. Estimating the Directed Spectrum from a single vector timeseries sample that was 1s in duration required 0.5-1.5 $s$ of wallclock time to compute. Each non-negative matrix factorization model in the simulated data experiment required approximately 1 minute of wallclock time to train. The NMF models in the local field potential experiment required between 2-20 minutes to train.

## G   The Directed Spectrum is robust to violations of its model assumptions

In order to evaluate whether the Directed Spectrum is robust to violations of Assumptions 1 and 2, we repeated the experiments described in Section 5 with alterations to the data simulation process. We first tested the impact of violations to Assumption 1 by adjusting the underlying VAR models so that the innovations within a given channel are correlated across all latent networks. We simulated five different covariances between networks, ranging from 0 (no correlation) to 1. (A covariance of 1 is the maximum possible covariance between channels within a network).

The Spearman's correlations between the true and estimated network scores for each model are reported in Table S2. At all covariance levels, the DS models performed better than every comparison method by a large margin. As expected, violations of Assumption 1 reduced the efficacy of the DS+LFM approach for recovering the true latent networks. But this reduction was not as severe as might be expected, with the DS model at a covariance of 1 still outperforming the best comparison method. Hence, violations of Assumption 1 do not change our recommendation of the Directed Spectrum as the optimal metric for use with LFMs to model directed communication in latent brain networks.

We performed two additional tests to measure the impact of violations to Assumption 2 on latent network recovery. In the first test, a cube root function ($x^{1/3}$) was applied to the raw timeseries data before calculating directed communication features, causing timeseries values to have a non-linear relationship with the underlying networks. This function was chosen because it is a nonlinear function that maintains sign, is monotonic, and is concave for positive values. We also tested mixtures of the original and nonlinear timeseries to measure the impact of weaker levels of nonlinearity. In the second test, we subtracted products between each of the network outputs from the original linear timeseries to produce nonlinear data. This results in a time series that only has nonlinear relationships to the networks when more than one network is active. The products between each network output were scaled by a factor $\lambda$ to observe the impact of this type of nonlinearity at different intensities. In this way, the timeseries data (Y) would be represented as

$$y = x_1 + x_2 + x_3 - \lambda(x_1 x_2 + x_2 x_3 + x_3 x_1), \tag{52}$$

where $x_i$ is the output of the $i^{th}$ network.

Tables S3 and S4 present the Spearman correlation between true and estimated latent network scores for these datasets. As expected, violations of Assumption 2 reduce the accuracy of model recovery. But in both tests, we see that the directed spectrum models outperform all comparison methods at

Table S2: Spearman's correlation between latent network activation estimates and true activation scores when Assumption 1 is violated. Each column is associated with a different covariance for innovations between networks. GC: unconditional Granger causality; PSI: phase slope index; DTF: directed transfer function; PDC: partial directed coherence; DS: Directed Spectrum.

| Covariance | 0.00 | 0.25 | 0.50 | 0.75 | 1.00 |
|---|---|---|---|---|---|
| GC *Network 1* | 0.511 | 0.179 | 0.130 | 0.051 | 0.102 |
| GC *Network 2* | 0.205 | 0.138 | 0.130 | 0.100 | 0.043 |
| GC *Network 3* | 0.123 | 0.195 | -0.073 | 0.015 | -0.055 |
| PSI *Network 1* | 0.405 | 0.074 | -0.005 | -0.035 | 0.001 |
| PSI *Network 2* | 0.170 | -0.059 | 0.026 | 0.028 | 0.026 |
| PSI *Network 3* | 0.238 | 0.165 | 0.128 | 0.099 | 0.063 |
| DTF *Network 1* | 0.614 | 0.478 | 0.321 | 0.243 | 0.167 |
| DTF *Network 2* | 0.371 | 0.082 | 0.035 | 0.008 | 0.009 |
| DTF *Network 3* | 0.265 | 0.138 | 0.233 | 0.122 | 0.077 |
| PDC *Network 1* | 0.590 | 0.333 | 0.068 | 0.030 | 0.095 |
| PDC *Network 2* | 0.382 | 0.017 | -0.000 | 0.002 | 0.006 |
| PDC *Network 3* | 0.245 | 0.335 | 0.254 | 0.199 | 0.176 |
| **DS** *Network 1* | 0.952 | 0.714 | 0.679 | 0.769 | 0.702 |
| **DS** *Network 2* | 0.933 | 0.436 | 0.322 | 0.417 | 0.373 |
| **DS** *Network 3* | 0.932 | 0.637 | 0.607 | 0.725 | 0.671 |

all levels of non-linearity. These results further justify the directed spectrum as the best available method for integrating measures of directed communication into LFMs.

Table S3: Spearman's correlation between latent network activation estimates and true activation scores when Assumption 2 is violated by applying a $x^{1/3}$ transform. Each column is associated with a different level of mixture between the linear and 3rd root data. A mixture value of 0 represents the orignal data; 1 represents the 3rd root data. GC: unconditional Granger causality; PSI: phase slope index; DTF: directed transfer function; PDC: partial directed coherence; DS: Directed Spectrum.

| Mix. Level | 0.00 | 0.25 | 0.50 | 0.75 | 1.00 |
|---|---|---|---|---|---|
| GC *Network 1* | 0.513 | 0.395 | 0.418 | 0.378 | 0.293 |
| GC *Network 2* | 0.499 | 0.401 | 0.461 | 0.541 | 0.597 |
| GC *Network 3* | 0.275 | 0.097 | 0.190 | 0.244 | 0.267 |
| PSI *Network 1* | 0.387 | 0.389 | 0.396 | 0.412 | 0.431 |
| PSI *Network 2* | 0.136 | 0.224 | 0.365 | 0.440 | 0.459 |
| PSI *Network 3* | 0.252 | 0.269 | 0.300 | 0.339 | 0.348 |
| DTF *Network 1* | 0.596 | 0.633 | 0.537 | 0.520 | 0.531 |
| DTF *Network 2* | 0.253 | 0.572 | 0.416 | 0.423 | 0.430 |
| DTF *Network 3* | 0.427 | 0.467 | 0.242 | 0.226 | 0.219 |
| PDC *Network 1* | 0.329 | 0.619 | 0.539 | 0.517 | 0.522 |
| PDC *Network 2* | 0.234 | 0.556 | 0.384 | 0.385 | 0.388 |
| PDC *Network 3* | 0.510 | 0.453 | 0.233 | 0.196 | 0.177 |
| **DS** *Network 1* | 0.949 | 0.919 | 0.871 | 0.828 | 0.800 |
| **DS** *Network 2* | 0.934 | 0.913 | 0.862 | 0.813 | 0.762 |
| **DS** *Network 3* | 0.936 | 0.893 | 0.802 | 0.661 | 0.734 |

## H  The Directed Spectrum is robust to shorter window lengths

In order to investigate the impact of window length on the results reported in Section 5, we repeated those experiments with different simulated recording window lengths. Window lengths of 0.5, 1, 2,

Table S4: Spearman's correlation between latent network activation estimates and true activation scores when Assumption 2 is violated by subtracting pairwise products of the network outputs from the original data. GC: unconditional Granger causality; PSI: phase slope index; DTF: directed transfer function; PDC: partial directed coherence; DS: Directed Spectrum.

| Scale Factor $\lambda$ | 0.00 | 0.25 | 0.50 | 0.75 | 1.00 |
|---|---|---|---|---|---|
| GC *Network 1* | 0.515 | 0.109 | 0.055 | 0.012 | 0.000 |
| GC *Network 2* | 0.516 | 0.503 | 0.480 | 0.292 | 0.240 |
| GC *Network 3* | 0.340 | 0.003 | 0.056 | 0.064 | 0.037 |
| PSI *Network 1* | 0.387 | 0.057 | -0.011 | -0.002 | 0.001 |
| PSI *Network 2* | 0.136 | 0.093 | 0.039 | 0.039 | 0.030 |
| PSI *Network 3* | 0.252 | 0.009 | 0.030 | 0.028 | 0.028 |
| DTF *Network 1* | 0.596 | 0.589 | 0.351 | 0.337 | 0.341 |
| DTF *Network 2* | 0.255 | 0.093 | 0.039 | 0.030 | 0.026 |
| DTF *Network 3* | 0.429 | 0.481 | 0.447 | 0.428 | 0.424 |
| PDC *Network 1* | 0.327 | 0.368 | 0.588 | 0.585 | 0.583 |
| PDC *Network 2* | 0.233 | 0.036 | 0.145 | 0.146 | 0.146 |
| PDC *Network 3* | 0.508 | 0.466 | 0.136 | 0.199 | 0.256 |
| **DS** *Network 1* | 0.949 | 0.674 | 0.705 | 0.693 | 0.690 |
| **DS** *Network 2* | 0.934 | 0.620 | 0.649 | 0.648 | 0.651 |
| **DS** *Network 3* | 0.936 | 0.610 | 0.707 | 0.682 | 0.676 |

10, and 20 seconds were each used to simulate 1000 independent recordings. DS and comparison measures were calculated for each set of simulated recordings, and those data were used as inputs to a non-negative matrix factorization model as described in Section 5. The correlations between the estimated and true network scores are reported for each window length in Table S5. We found that estimates of phase slope index, directed transfer function, and partial directed coherence suffered from numerical instability issues at the shortest windows lengths; those data are omitted from the table. Increasing the window length improves network recovery for models based on both Directed Spectrum measures. At all window lengths the DS model performs markedly better than the comparison models.

## I Fourier transform comparison

In applications where the primary goal is decoding accuracy and characterizing directed communication within networks is unimportant, there are a number of undirected measures that could be used in a LFM to model latent brain networks. One of the most straightforward of these is to simply represent data in the frequency domain via a Fourier transform. In order to evaluate whether the additional effort of calculating the Directed Spectrum is worth it in such cases, we repeated the modeling procedures outlined in Sections 5 and 6 with the absolute value of the Fourier transform of the raw timeseries data as an additional set of comparison measures. Those results are reported as "FFT" in Tables S6 and S7, along with the already reported performance of the DS models. The FFT models did not perform significantly better or worse than the DS models. This is unsurprising. Given our assumptions, the raw FFT features are also linear functions of brain networks and may still work perfectly well in applications where the identification of directed communication between brain regions in latent networks is not desired.

## J Potential negative societal impacts

We are not aware of major concerns in the brain network modeling applications described in this work, as most human brain recordings occur in relatively secure healthcare settings and such data is likely of limited value to nefarious actors. If pressed, the main long-term area where this work could have negative impacts is in privacy. If at some point internet-connected non-invasive consumer-grade brain recording devices become more common, it is possible that the models described here could be used by undesired third parties to obtain information about an individual's neural state. Outside of

Table S5: Spearman's correlation between latent network activation estimates and true activation scores for varied window lengths. *10000 samples were generated for 5s windows; 1000 samples were generated for all other window lengths. Measure estimates for phase slope index, directed transfer function, and partial directed coherence suffered from stability issues for window lengths of 0.5 s. GC: unconditional Granger causality; PSI: phase slope index; DTF: directed transfer function; PDC: partial directed coherence; DS: Directed Spectrum; PDS: Pairwise Directed Spectrum.

| Window Length | 0.5 s | 1 s | 2 s | 5 s* | 10 s | 20 s |
|---|---|---|---|---|---|---|
| GC *Network 1* | 0.089 | 0.113 | 0.457 | 0.485 | 0.510 | 0.685 |
| GC *Network 2* | 0.345 | 0.419 | 0.487 | 0.442 | 0.443 | 0.585 |
| GC *Network 3* | 0.090 | 0.167 | 0.426 | 0.281 | 0.200 | 0.509 |
| PSI *Network 1* | - | 0.028 | 0.162 | 0.387 | 0.509 | 0.600 |
| PSI *Network 2* | - | 0.014 | 0.020 | 0.135 | 0.232 | 0.253 |
| PSI *Network 3* | - | 0.013 | 0.138 | 0.248 | 0.549 | 0.581 |
| DTF *Network 1* | - | 0.550 | 0.535 | 0.426 | 0.707 | 0.570 |
| DTF *Network 2* | - | 0.151 | 0.173 | 0.131 | 0.608 | 0.318 |
| DTF *Network 3* | - | 0.459 | 0.263 | 0.542 | 0.612 | 0.364 |
| PDC *Network 1* | - | 0.359 | 0.329 | 0.560 | 0.474 | 0.563 |
| PDC *Network 2* | - | 0.138 | 0.131 | 0.154 | 0.186 | 0.304 |
| PDC *Network 3* | - | 0.534 | 0.541 | 0.445 | 0.478 | 0.410 |
| **DS** *Network 1* | 0.763 | 0.886 | 0.929 | 0.920 | 0.958 | 0.964 |
| **DS** *Network 2* | 0.602 | 0.868 | 0.907 | 0.905 | 0.944 | 0.945 |
| **DS** *Network 3* | 0.449 | 0.897 | 0.921 | 0.927 | 0.932 | 0.928 |
| **PDS** *Network 1* | 0.803 | 0.863 | 0.883 | 0.908 | 0.920 | 0.938 |
| **PDS** *Network 2* | 0.728 | 0.879 | 0.891 | 0.917 | 0.925 | 0.925 |
| **PDS** *Network 3* | 0.796 | 0.857 | 0.888 | 0.916 | 0.917 | 0.915 |

Table S6: Spearman's correlation between FFT model latent network activation estimates and true activation scores. FFT: fast Fourier transform; DS: directed spectrum; PDS: pairwise directed spectrum. Values in *[brackets]* represent the 95% confidence interval [45].

| Measure | Network 1 | Network 2 | Network 3 |
|---|---|---|---|
| FFT | 0.889 *[0.883, 0.893]* | 0.939 *[0.936, 0.941]* | 0.921 *[0.917, 0.924]* |
| **DS** | 0.920 *[0.916, 0.923]* | 0.905 *[0.901, 0.909]* | 0.927 *[0.924, 0.930]* |
| **PDS** | 0.908 *[0.904, 0.912]* | 0.917 *[0.913, 0.920]* | 0.916 *[0.913, 0.920]* |

that potential future possibility, there may be other applications beyond the scope of this work where the ability to identify latent networks states could lead to privacy concerns. These concerns, at this stage, appear minimal and are not unique to this work.

Table S7: Behavioral Context Decoding Performance for FFT models. The columns 'HC AUC', 'OF AUC', and 'TS AUC' report the mean and standard error of the one-vs-all AUC across 5 splits for the homecage, open field, and tail suspension behavioral contexts, respectively. The 'Mean AUC' column reports the average across the mean AUCs reported for each behavioral context.

| Measure | *Mean AUC* | HC AUC | OF AUC | TS AUC |
|---------|-----------|--------|--------|--------|
| FFT | 0.913 | $0.893 \pm 0.018$ | $0.922 \pm 0.014$ | $0.924 \pm 0.006$ |
| **DS** | 0.908 | $0.894 \pm 0.016$ | $0.916 \pm 0.012$ | $0.915 \pm 0.007$ |
| **PDS** | 0.919 | $0.909 \pm 0.014$ | $0.915 \pm 0.014$ | $0.932 \pm 0.005$ |