# OpenReview forum: "Directed Spectrum Measures Improve Latent Network Models Of Neural Populations"
_NeurIPS.cc/2021/Conference — NeurIPS 2021 Poster_

### Official Review · Reviewer_3jsZ · 2021-07-16

**Rating:** 7
**Confidence:** 3

**Summary:**

The authors define the Directed Spectrum (DS) of multivariate timeseries data. It is shown that under certain conditions the DS is a linear function of latent brain networks. In synthetic data, the DS (and NMF) was better able to recover ground truth networks compared to a set of other directed measures. In neural data from mice, the network activations derived from DS are better able to predict behavioral state.

**Limitations And Societal Impact:**

The authors address limitations of the DS based on its assumptions of stationarity and its use with LLFMs. It would be good to know how sensitive the method is to violations of Assumptions 1 or 2 as they likely don't hold exactly in neural data. If there are weak correlations in the innovations or if the transmitted signals combine sub-additively, how quickly do the estimates break down?

**Main Review:**

The motivation for the manuscript, careful consideration of the interplay between LLFMs and measures of neural data, and results were all clear. The DS is well motivated and is put into the context of other directed measures. I have one broad suggestion in terms of the presentation of Section 3 and some minor questions and suggestions.

In my opinion, the main weakness  is that Section 3 is very dense and relies on a number of definitions and results that are only in the appendix, which made it difficult to read through as a self-contained section. I think it would benefit the clarity of Section 3 to walk the reader through some definitions (e.g., how to interpret $H^{(j)}(\omega)$) in Eq 5 and the interpretations of the terms in certain definitions (e.g., Eq 9). Some of these are in the appendix, but I would suggest using some of the additional page to make this section more self-contained.

- I found that using $x$ for both the data vector and latent factors in Eq 1 to be confusing. I would suggest changing the latent factor variable.
- In lines 131-134, I think it would be helpful to have more exposition about when it could be useful to define groups versus single channels.
- For Section 6, is it possible to better differentiate whether the DS is better for decoding as an engineered features versus whether some small subset of the inferred networks are better for decoding. Since there are only 3 behavioral contexts, it would be interesting if a small set of network activations were responsible for decoding versus if many networks are being used.
- Should equations 12 and 13 be part of one multiline equation?
- Line 278: L1 factorization -> L1 regularization

**Time Spent Reviewing:**

5

---

> ### Author Response · Authors · 2021-08-09
> **Review 3jsZ Response**
>
> >'The motivation for the manuscript, careful consideration of the interplay between LLFMs and measures of neural data, and results were all clear. The DS is well motivated and is put into the context of other directed measures.'
>
> We really appreciate that you thought the directed spectrum metric was well motivated and that most of the manuscript was clear.
>
> ---
>
> >'I have one broad suggestion in terms of the presentation of Section 3 and some minor questions and suggestions. In my opinion, the main weakness is that Section 3 is very dense and relies on a number of definitions and results that are only in the appendix, which made it difficult to read through as a self-contained section.'
>
> With regards to the clarity of section 3 and the other suggestions, your comments here are helpful and we plan to take them into consideration when making final edits for the manuscript. If you have more specific suggestions regarding how to make specific terms or definitions more clear, we would appreciate hearing them.
>
> ---
>
> >'For Section 6, is it possible to better differentiate whether the DS is better for decoding as an engineered features versus whether some small subset of the inferred networks are better for decoding.'
>
> This is an excellent question. In a larger collaborative project where we are using the DS features and LLFMs to decode psychiatric states, we have found that a single network from the larger model is sufficient to achieve good decoding performance with minimal gains from using additional networks. We suspect that the same is true here, and that only 2 or 3 networks are necessary to achieve good decoding performance.

---

> > ### Comment · Reviewer_3jsZ · 2021-08-17
> > **Response to authors**
> >
> > Thanks for the reply which addresses my comments.

---

### Official Review · Reviewer_mATF · 2021-07-16

**Rating:** 6
**Confidence:** 4

**Summary:**

The authors present a method for estimating the directional influences between channels in order estimate the existence and structure of distinct, latent networks. They call their metric the "directed spectrum" and demonstrate its superiority over Granger causality to estimate directed influences between recording channels. The motivation for their approach is the paucity of directed influence metrics that operate in conjunction with latent variable models of neural population activity.

**Limitations And Societal Impact:**

yes.

**Main Review:**

The paper is well written and flows nicely. The metric itself is fairly simple and it would be surprising if it didn't already exist under some other name. Although the authors appear to know and cite the relevant literature for this topic. One point of concern may be that the metric could have unfavorable sampling properties. For example, it is well known that common estimators for the cross-spectral matrix may be either inconsistent and/or high-variance. Does their estimator inherit these properties? Also, could the authors comment on how the metric might be influenced by model mismatch (eg.  generative models with non-linear dynamics, Poisson measurements).

While I don't mind some of their basic assumptions I find the arguments about innovations and independence (in particular in lines 148-153) somewhat weak. It is not at all obvious that brain networks wouldn't want to bounce existing information back-and-forth. The authors would have to substantiate this claim more throughly. That said, I would prefer to accept that this is a basic assumption of the model and being clear-eyed about how that may impact results rather than swallow the claim that somehow the brain prefers to operate this way.

I must admit that I am dubious of directed measures of neural population activity in general as I find it difficult to imagine a brain with recurrent connections showing anything other than directed activity in both directions. For example, the LFP data the authors present in the Supplement seems to indicate that none of the other brain areas direct their activity to the nucleus accumbens. Would we not expect VTA to act upon NAc? That does not appear to be the case based on the DS measure. How would the authors have us understand this inconsistency?

Lastly, I'm glad that the authors have explored this method and it's great that it performs so well on the classification task but I'm curious about the authors' thoughts on what new science can be done with this approach. Is this merely a tool for functional network discovery or did they have something else in mind?

As an aside, the authors may be interested in some recent work by Yousefi et al. (2019) describing the "global coherence" in which the authors develop an explicit latent variable model for network activity that is also designed to learn the identity of latent networks.

A. Yousefi, R. S. Fard, U. T. Eden and E. N. Brown, "State-Space Global Coherence to Estimate the Spatio-Temporal Dynamics of the Coordinated Brain Activity," 2019 41st Annual International Conference of the IEEE Engineering in Medicine and Biology Society (EMBC), 2019, pp. 5794-5798, doi: 10.1109/EMBC.2019.8856634.

**Time Spent Reviewing:**

3

---

> ### Author Response · Authors · 2021-08-09
> **Review mATF Response**
>
> >'The metric itself is fairly simple and it would be surprising if it didn't already exist under some other name.'
>
> The metric that we have named the Directed Spectrum is explicitly mentioned by Geweke in his 1982 paper introducing spectral Granger causality (see Eqn. 3.5 in that paper). However, Geweke only uses it as an intermediary for deriving spectral Granger causality, and does not give it a name. Beyond that we are not aware any further work exploring this metric.
>
> ---
>
> >'For example, it is well known that common estimators for the cross-spectral matrix may be either inconsistent and/or high-variance. Does their estimator inherit these properties?'
>
> This is an interesting question that we don't fully know the answer to. Now that you have brought it to our attention, we would like to do a more thorough evaluation. In practice, we have found that our method is sufficiently robust when using Welch's method for cross-spectral matrix estimation.
>
> ---
>
> >'While I don't mind some of their basic assumptions I find the arguments about innovations and independence (in particular in lines 148-153) somewhat weak.'
>
> Thank you for pointing out the weaknesses here. It seems that the claims in lines 148-153 are worded more strongly than we intended, and we will adjust those lines accordingly. Our intention was for the definition of TS to simply be a basic assumption of the model (rather than trying to argue that we know this is how the brain operates). Lines 148-153 are primarily meant to explain to the reader why this particular assumption was chosen over other assumptions that could have been made. Another argument that biased us towards this particular definition of TS was that at latencies lower than the speed of action potential propagation between regions, the assumption in (9) is the only one that makes physical sense (and having different assumptions for different latencies seemed too complicated).
>
> ---
>
> >'For example, the LFP data the authors present in the Supplement seems to indicate that none of the other brain areas direct their activity to the nucleus accumbens.'
>
> We apologize if the presentation of Figure S2 and its relationship to the corresponding model was unclear. This figure represents a single factor from the NMF model (i.e. a single network). At any given point in a recording, the model will typically indicate that many different networks are active simultaneously, some of which may contain activity being directed towards nucleus accumbens from other regions.  We will add visualizations of additional networks to the supplement, which will make it clear that this connection does appear in other networks.
>
> Additionally, even for the single network plotted in Figure S2, it is highly possible that the firing of individual cells that propagate from VTA to NAc is correlated with activation of this network. In our experience the local field potential data we are using here typically only picks up on activity that is synchronized across a large population of neurons. Thus, any neural activity that is not synchronized with a relatively significant portion of the population in a region may be missed in this way. Please let us know if you think further discussion of this point is warranted in the manuscript.
>
> ---
>
> >'Is this merely a tool for functional network discovery or did they have something else in mind?'
>
> While the main goal of this manuscript is to demonstrate the Directed Spectrum and LLFMs as a tool for functional network discovery, we do believe there is potential for other scientific applications. For example, we plan to use Directed Spectrum estimates to provide feedback in a system designed for closed-loop control of functional network states. In addition, we believe that information from functional network models that incorporate the Directed Spectrum can be used to choose appropriate locations for neural stimulation to control functional network states even in open-loop stimulation scenarios.
>
> ---
>
> Finally, thank you for bringing the work by Yousefi et al. to our attention. It is an interesting method and worth referencing in our manuscript as a recent development that is complementary to ours for closely related scientific applications.

---

### Official Review · Reviewer_pzTq · 2021-07-16

**Rating:** 6
**Confidence:** 4

**Summary:**

This study introduces a new metric (Directed Spectrum or DS) that is compatible with linear latent factor methods (LLF) and, therefore, can be used for latent network discovery with such methods. First, the authors show that there exists a linear relationship between the DS between channels and the DS in latent networks. Using simulated data, the authors estimate latent networks by applying NMF directly to the DS estimates. They show stronger correlations with latent activations recovered by their method, as compared to other traditional metrics like spectral GC, PDC, DTF, and the like. A specific comparison reveals that GC estimates far more spurious connections as compared to the DS metric. Finally, the authors also show that, in a neuroscience dataset, the latents estimated by DS features better predict the behavioral states as compared to latents estimated by traditional metrics.

**Ethical Concerns:**

No.

**Limitations And Societal Impact:**

Yes.

**Main Review:**

Preamble: I am quite familiar with measures of directed communication applied in neuroscience, and routinely employ these with our research data. But I am not an expert on the theoretical underpinnings of these measures. I will, therefore, append each critique with a confidence score (10/highest to 1/lowest), to enable the authors to optimally address my criticisms in the rebuttal, considering space constraints.

Originality: 7/10

Quality: 7/10

Clarity: 6/10

Significance: 6/10

a) Regarding the comparison with spectral GC:

i) There is a particular challenge with GC when inferring causality because the GC often estimates weights for both forward and (spurious) backward connections. In this case, differencing the estimates has been found to provide some control over the spurious backward connections (Roebroeck, Formisano, Goebel, 2005). I would suggest the authors use this metric (Fy->x(w) - Fx->y(w)) for their comparison rather than GC per se.

ii) Another aspect that was unclear from the main paper was whether the authors used pairwise GC or conditional GC, although both have been defined in Supplemental section C.3. If they had used pairwise GC, could they also provide results with conditional GC (or both)?

iii) Is it possible that GC performs better with data from longer time windows? Fig 3 could be plotted for different lengths of windows (e.g. with mean and errorbors, rather than showing the estimates for every window/sample).

*Confidence: 10


b) On a related note. Please show the directed influence spectra (analog of Fig.4) with the other approaches also, at least in summary form. Did these other approaches perform comparably with DS? The proportion of spurious connections estimated with all approaches should be quantified for comparison.

*Confidence: 10


c) (line 160) The authors indicate that "Assumption 1 can be thought of as enforcing the independence of inputs to each network". How valid is this assumption in a neurobiological setting? Of course, some components of the innovations may reflect "independent noise", but visual or other sensory inputs are likely to be highly correlated among networks (e.g. the LGN projects to multiple high level visual areas, in parallel). Can the authors clarify, and perhaps explore to what level the model is tolerant to a failure of this assumption?

*Confidence: 10


d) The behavioral context decoding section needs to be updated significantly. One would need to see what accuracies can be obtained simply from the "raw" timeseries data or from simple spectral features extracted from these data, for each context. Having established this baseline, it would then make sense to show that DS features outperform these simpler features in terms of decoding behavioral state. Otherwise, the results could be simply explained by DS+NMF producing the least "distortion" in the raw data (given that the other methods are non-linear).

*Confidence: 10


e) The authors seek to provide a thorough and detailed description, but some equations and derivations could be made clearer, at least in the Supporting Information. For example, the "transmitted signal" (based on the conditional density) introduced in equation 9 -- which is at the heart of the model -- could be derived in the supporting information with a clear explanation of how each term maps on to the stated assumptions in the immediately preceding paragraph.

*Confidence: 8


f) Given that the main contribution of the paper seems to be theoretical, the precise relationship between DS and traditional metrics should be clearly described, at least in the Supplemental information. Such a derivation would ideally follow the Supplemental Section E on the relationship with the cross-spectral matrix and its factorization. While Supplemental section C presents the equations associated with other metrics, such a description would allow the reader to also understand the relative advantages and limitations of the DS metrics vis-a-vis traditional metrics.

*Confidence: 7


g) On a related note, a few more empirical simulations could be presented that explore the limits of the proposed metric, and its tolerance to violations of key assumptions. For example, what happens when latents mix non-linearly. Or network interactions characterized by operations other than scaling and delay? In the brain, it is known that the phase of slow oscillations modulates the amplitude of faster oscillations. Could these types of interactions also be captured by DS, and better than DTF or GC?

*Confidence: 7


h) As an empirical control it would be nice to see a baseline comparison with latent linear factor models directly applied to the time series data, followed by directed influence estimation with DS or other traditional methods. In the case of DS I guess the order will not matter significantly, given linearity assumptions. But how do the other methods fare when NMF is applied to the data followed by, say, GC estimation on the latents compared to NMF on the channel-wise GC estimates? Are there certain interactions that can be recovered only by the latter method vs the former?

*Confidence: 6


i) I am not sure the sample subscript 'n' adds anything that cannot be understood without it. It shows up in every term and clutters up the equations, and gets dropped by mistake occasionally (Supplemental Section S1). I'd suggest removing it.

*Confidence: 6


**Time Spent Reviewing:**

5

---

> ### Author Response · Authors · 2021-08-09
> **Review pzTq Response**
>
> Thank you for labeling each of the topics in your review. Our responses to each topic are given with the corresponding labels below.
>
> a.i) We appreciate that you brought the Roebroeck, Formisano, Goebel (2005) paper to our attention. Given that the influence difference metric you suggest appears to be more common in fMRI applications, we plan to add it as an additional comparison metric. Based on the reference you provided, the benefits of the influence difference over standard Granger causality are due to the BOLD response function and slow temporal sampling of fMRI data, so it is unclear whether we would see an increase or a decrease in performance when applying it to our LFP dataset. We note that the paper presents LFP data as a 'ground truth' scenario where the influence difference is not necessary to compensate for the BOLD response and slow sampling.
>
> a.ii) The results we present use standard (pairwise) Granger causality. We will repeat all of the Granger causality calculations with conditional Granger causality to address your concern. However, we do not expect a substantial change in performance, since most of the spurious connections are not 'passthrough' connections (e.g. when A>C is inferred because A>B>C is true).
>
> a.iii) Based on our own testing, we found the window lengths used in the simulated data experiment were sufficient for stable Granger causality estimates. These results were not presented in the original manuscript, and we will add details on these evaluations to the supplement.
>
> b) Plotting the estimated factors for all comparison methods is a good idea. We can include these results in the supplement. We singled out Granger causality to show in the original manuscript because it appeared to give the best factor estimation of all the comparison methods.
>
> Quantifying the reconstruction performance is more challenging because there is not a clear way to compare what the “correct” reconstruction should be for each metric since they are all different. That is why we only focused on comparing score estimation, which allows for a more objective comparison.
>
> c) The point that we should expect some inputs (like sensory input) to be correlated between network is astute. Assumption 1 was a necessary oversimplification to make our model mathematically tractable. We believe that the models we describe should be relatively tolerant to violations of this assumption, and additional simulations on this issue will be added. (See common response).
>
> d) This is a reasonable request, and we will add this as 'sanity check' comparison method. From previous experience, we expect that the directed spectrum will still outperform an NMF model applied to the raw data. It is also worth noting that this will not be informative about directed communication within functional networks, and in our scientific application is not a ‘competing’ method.
>
> e) Thank you for pointing this out. It would be helpful to hear more details about sections or equations that were particularly unclear so that we can improve them.
>
> f) We appreciate your interest in the Directed Spectrum in this regard and can add additional information to supplemental section C relating DS to the other VAR-based metrics. While we are interested in a more full comparison of the benefits and advantages of metrics for directed communication, that is outside the scope of this work where we are focusing on the compatibility of such metrics with LLFMs.
>
> g) Measuring how these models perform when the data are generated in a way that does not match the model assumptions is a good idea (see common response).
>
> It is not straightforward how amplitude modulation would be represented by the Directed Spectrum. If you think it would improve the manuscript, we could include some examples in the supplement to show what this looks like empirically.
>
> h) Could you elaborate on the scientific question that this control is meant to address? One challenge here is that NMF cannot be applied to the timeseries LFP data because it contains negative values.
>
> i) Thanks for pointing out that the purpose of the subscript 'n' was unclear.  I believe all the places where it is dropped are intentional. Those variables without the 'n' subscript indicate values that remain the same over all samples. Variables with an 'n' subscript differ from sample to sample.

---

> > ### Comment · Reviewer_pzTq · 2021-08-25
> > **response**
> >
> > Thank you for your responses.
> >
> > a.i) the difference of influence (DOI) term is relevant when one expects smoothing of the raw signals. This may be relevant for filtered LFPs also and has been used previously with ECoG data (Bastin et al, 2016, Cereb. Cortex).
> >
> > d) it would be helpful to see these numbers to confirm the validity of the claims.
> >
> > e) please see reviewer 3jsZ's comments also ("I think it would benefit the clarity of Section 3 to walk the reader through some definitions")

---

> > > ### Author Response · Authors · 2021-09-07
> > > **d)**
> > >
> > > The main goal of the Directed Spectrum is to provide a simple method for estimating directed communication within latent brain networks. With regards to d), we emphasize that using fft or other ‘raw’ features is a complementary, rather than competing, method that achieves a different goal. Based on your request, we revisited our earlier comparisons and found some interesting results. When we use the absolute value of the Fourier transform of the raw data as features in an NMF model with a mean-square error loss and no regularization, we saw worse reconstruction of the simulated networks described in section 5 compared to a similar NMF model using Directed Spectrum features. The following table provides Spearman correlations between the true and estimated network activations for both models:
> > >
> > > ---
> > >  | Network 1 | Network 2| Network3 |
> > >
> > > ---
> > >
> > > *DS* | 0.89603 | 0.89326 | 0.86119 |
> > >
> > > *FFT* | 0.78354 | 0.76880 | 0.75501 |
> > >
> > > ---
> > >
> > > If we swap out the NMF model for one with an Itakura-Saito divergence loss and L1 regularization (matching what was used to generate the results in section 5) we find that using the fft features performs similarly to the Directed Spectrum features.
> > >
> > > ---
> > >  | Network 1 | Network 2| Network3 |
> > >
> > > ---
> > >
> > > *DS* | 0.91950 | 0.90489 | 0.92720 |
> > >
> > > *FFT* | 0.88786 | 0.93866 | 0.92067 |
> > >
> > > ---
> > > These results suggest you are right that the Directed Spectrum models perform well, at least in part, because they produce the least 'distortion' of the raw signal. In hindsight, this makes sense and matches the fact that our motivation for the Directed Spectrum was to provide a measure of directed communication that was a linear function of underlying brain networks. Given our assumptions, the raw fft features are also linear functions of brain networks and may still work perfectly well in applications where the identification of directed communication between brain regions in latent networks is not desired. (Although our results emphasize that the Itakura-Saito divergence loss should be used in this case rather than the typical MSE loss for NMF models.)
> > >
> > > We could, of course, engineer situations that that explicitly require directed features to recover the networks.  For example, consider 2 networks that are made up of a unidirectional signal between two regions, A->B or B->A, in the same frequency band. These two networks would be recoverable by using directed features, whereas fft features would not be able to recover the networks in this case.

---

### Author Response · Authors · 2021-08-09
**Common Response to All Reviews**

We would like to thank both the reviewers and the area chairs for the time they've put into evaluating our work. Each of the reviewers provided a good summary of our work as well as useful insights into ways that we can make the work even stronger. We have provided separate responses for comments in each of the individual reviews. One theme that was common to all the reviews is addressed here, rather than in those individual responses.

### Reviewer comments
**mATF**: 'could the authors comment on how the metric might be influenced by model mismatch'
**pzTq**: 'a few more empirical simulations could be presented that explore the limits of the proposed metric, and its tolerance to violations of key assumptions'
**3jsZ**: 'It would be good to know how sensitive the method is to violations of Assumptions 1 or 2 as they likely don't hold exactly in neural data.'
### Response
Many violations of our model assumptions (including Assumptions 1 and 2) only impact how closely the Directed Spectrum adheres to a linear relationship with the underlying networks. If the assumptions are completely violated, it would correspond to a non-linear relationship similar to many of the discussed metrics (e.g., Granger Causality) and we would expect similar performance between the Directed Spectrum and other existing approaches.  In practice, we expect that small violation is reasonably handled by the noise terms of an LLFM model.
Because all 3 reviewers mentioned this as a concern, we plan to add an additional section to the supplement discussing this issue and showing how well the DS and LLFMs recover simulated network data that violates the assumed model.

---

> ### Author Response · Authors · 2021-08-18
> **Follow-up analysis**
>
> We conducted a preliminary test to measure the impact of violations to Assumption 1 on latent network recovery.  Specifically, we repeated the experiment in Section 5 with a single alteration to the way the data was generated.  The innovations of the underlying VAR model were adjusted so that the innovations within a given channel were correlated across all latent networks. We simulated five different covariances between networks, ranging from 0 (no correlation) to 1. (A covariance of 1 is the maximum possible covariance between channels within a network).
>
> At all covariance levels, the DS models performed better than every comparison method by a large margin. As expected, violations of Assumption 1 reduce the efficacy of the DS+LLFM approach for recovering the true latent networks. But this reduction was not as severe as we expected, with the DS model at a covariance of 1 still outperforming the best comparison method. Hence, violations of Assumption 1 do not change our recommendation of DS as the optimal metric for use with LLFMs to model directed communication in latent brain networks.
>
> The following table represents the Spearman correlation between the true and estimated latent network scores for each method and covariance:
>
> ---
> Covariance      | **0.000** | **0.250** | **0.500** | **0.750** | **1.000** |
>
> ---
>
> *DS* Network 1 | 0.952 | 0.714 | 0.679  | 0.769  | 0.702  |
>
> *DS* Network 2 | 0.933 | 0.436  | 0.322  | 0.417 | 0.373  |
>
> *DS* Network 3 | 0.932 | 0.637  | 0.607  | 0.725 | 0.671  |
>
> ---
>
> *GC* Network 1 | 0.511 | 0.179  | 0.130  | 0.051  | 0.102  |
>
> *GC* Network 2 | 0.205 | 0.138  | 0.130  | 0.100  | 0.043  |
>
> *GC* Network 3 | 0.123 | 0.195  | -0.073 | 0.015  | -0.055 |
>
> ---
>
> *PSI* Network 1 | 0.405 | 0.074  | -0.005  | -0.035  | 0.001  |
>
> *PSI* Network 2 | 0.170  | -0.059  | 0.026  | 0.028  | 0.026  |
>
> *PSI* Network 3 | 0.238 | 0.165  | 0.128  | 0.099  | 0.063 |
>
> ---
>
> *DTF* Network 1 | 0.614 | 0.478  | 0.321  | 0.243  | 0.167  |
>
> *DTF* Network 2 | 0.371 | 0.082  | 0.035  | 0.008  | 0.009  |
>
> *DTF* Network 3 | 0.265 | 0.138  | 0.233  | 0.122  | 0.077  |
>
> ---
>
> *PDC* Network 1 | 0.590 | 0.333  | 0.068  | 0.030  | 0.095  |
>
> *PDC* Network 2 | 0.382 | 0.017  | -0.000  | 0.002  | 0.006  |
>
> *PDC* Network 3 | 0.245 | 0.335  | 0.254  | 0.199  | 0.176 |
>
> ---
>
> We are in the process of testing the impact of other types of violations of the model assumptions on recovery of underlying networks.  These robustness analyses will be added in detail to the Supplemental Material.

---

> > ### Author Response · Authors · 2021-08-26
> > **Follow-up pt. 2**
> >
> > We performed two additional tests to measure the impact of violations to Assumption 2 on latent network recovery.
> > In the first test, a cube root function ($x^{1/3}$) was applied to the raw timeseries data before calculating directed communication features, causing timeseries values to have a non-linear relationship with the underlying networks.
> > This function was chosen because it is a nonlinear function that maintains sign, is monotonic, and is concave for positive values.
> > We also tested mixtures of the original and nonlinear timeseries to measure the impact of weaker levels of nonlinearity.
> > The following table presents the Spearman correlation between true and estimated latent network scores for this dataset (a mixture value of 0 represents the orignal data; 1 represents the 3rd root data):
> >
> > ---
> > Nonlin. Mixture | **0.000** |**0.250** | **0.500** | **0.750** | **1.000** |
> >
> > ---
> >
> > *DS* Network 1 | 0.949 | 0.919 | 0.871 | 0.828 | 0.800 |
> >
> > *DS* Network 2 | 0.934 | 0.913 | 0.862 | 0.813 | 0.762 |
> >
> > *DS* Network 3 | 0.936 | 0.893 | 0.802 | 0.661 | 0.734 |
> >
> > ---
> >
> > *GC* Network 1 | 0.513 | 0.395 | 0.418 | 0.378 | 0.293 |
> >
> > *GC* Network 2 | 0.499 | 0.401 | 0.461 | 0.541 | 0.597 |
> >
> > *GC* Network 3 | 0.275 | 0.097 | 0.190 | 0.244 | 0.267 |
> >
> > ---
> >
> > *PSI* Network 1 | 0.387 | 0.389 | 0.396 | 0.412 | 0.431 |
> >
> > *PSI* Network 2 | 0.136 | 0.224 | 0.365 | 0.440 | 0.459 |
> >
> > *PSI* Network 3 | 0.252 | 0.269 | 0.300 | 0.339 | 0.348 |
> >
> > ---
> >
> > *DTF* Network 1 | 0.596 | 0.633 | 0.537 | 0.520 | 0.531 |
> >
> > *DTF* Network 2 | 0.253 | 0.572 | 0.416 | 0.423 | 0.430 |
> >
> > *DTF* Network 3 | 0.427 | 0.467 | 0.242 | 0.226 | 0.219 |
> >
> > ---
> >
> > *PDC* Network 1 | 0.329 | 0.619 | 0.539 | 0.517 | 0.522 |
> >
> > *PDC* Network 2 | 0.234 | 0.556 | 0.384 | 0.385 | 0.388 |
> >
> > *PDC* Network 3 | 0.510 | 0.453 | 0.233 | 0.196 | 0.177 |
> >
> > ---
> >
> > In the second test, we subtracted products between each of the network outputs from the original timeseries to produce data.
> > This results in a time series that only has nonlinear relationships to the networks when more than one network is active.
> > These products were scaled to observe the impact of this type of nonlinearity at different intensities.
> > In this way, the timeseries data (Y) would be represented as
> > $$y = x_1 + x_2 + x_3 - ScaleFactor*(x_1 * x_2 + x_2 * x_3 + x_3  * x_1) $$,
> > where $x_i$ is the output of the $i^{th}$ network.
> > The following table represents the Spearman correlations between the true and estimated latent network scores for this test:
> > ---
> >
> > Scale Factor | **0.000** |**0.250** | **0.500** | **0.750** | **1.000** |
> >
> > ---
> >
> > *DS* Network 1 | 0.949 | 0.674 | 0.705 | 0.693 | 0.690 |
> >
> > *DS* Network 2 | 0.934 | 0.620 | 0.649 | 0.648 | 0.651 |
> >
> > *DS* Network 3 | 0.936 | 0.610 | 0.707 | 0.682 | 0.676 |
> >
> > ---
> >
> > *GC* Network 1 | 0.515 | 0.109 | 0.055 | 0.012 | 0.000 |
> >
> > *GC* Network 2 | 0.516 | 0.503 | 0.480 | 0.292 | 0.240 |
> >
> > *GC* Network 3 | 0.340 | 0.003 | 0.056 | 0.064 | 0.037 |
> >
> > ---
> >
> > *PSI* Network 1 | 0.387 | 0.057 | -0.011 | -0.002 | 0.001 |
> >
> > *PSI* Network 2 | 0.136 | 0.093 | 0.039 | 0.039 | 0.030 |
> >
> > *PSI* Network 3 | 0.252 | 0.009 | 0.030 | 0.028 | 0.028 |
> >
> > ---
> >
> > *DTF* Network 1 | 0.596 | 0.589 | 0.351 | 0.337 | 0.341 |
> >
> > *DTF* Network 2 | 0.255 | 0.093 | 0.039 | 0.030 | 0.026 |
> >
> > *DTF* Network 3 | 0.429 | 0.481 | 0.447 | 0.428 | 0.424 |
> >
> > ---
> >
> > *PDC* Network 1 | 0.327 | 0.368 | 0.588 | 0.585 | 0.583 |
> >
> > *PDC* Network 2 | 0.233 | 0.036 | 0.145 | 0.146 | 0.146 |
> >
> > *PDC* Network 3 | 0.508 | 0.466 | 0.136 | 0.199 | 0.256 |
> >
> > ---
> >
> > As expected, violations of Assumption 2 reduce the accuracy of model recovery.
> > But in both tests, we see that the directed spectrum models outperform all comparison methods at all levels of non-linearity.
> > We believe that these results further justify the directed spectrum as the best available method for integrating measures of directed communication into LLFMs.

---

### Decision · Program_Chairs · 2021-09-27

**Decision:**

Accept (Poster)

**Comment:**

Through a productive discussion with the authors, the reviewers came to a borderline/accept consensus. The proposed "directed spectrum" measure is simple (in fact, the authors say it was an introduced as an intermediate quantity in a 1982 paper by Geweke), but appears to have some nice properties. While some reviewers praised the clarity of this paper, Reviewer 3jsZ and I found the paper rather dense (and the unnecessary subscripts didn't help).  I hope the authors will work to improve this aspect of the paper before publication.